# Credal Learning Theory

**Michele Caprio**
Department of Computer Science
University of Manchester, Manchester, UK
`michele.caprio@manchester.ac.uk`

**Maryam Sultana**    **Eleni G. Elia**    **Fabio Cuzzolin**
School of Engineering Computing & Mathematics
Oxford Brookes University, Oxford, UK
`{msultana,eelia,fabio.cuzzolin}@brookes.ac.uk`

## Abstract

Statistical learning theory is the foundation of machine learning, providing theoretical bounds for the risk of models learned from a (single) training set, assumed to issue from an unknown probability distribution. In actual deployment, however, the data distribution may (and often does) vary, causing domain adaptation/generalization issues. In this paper we lay the foundations for a 'credal' theory of learning, using convex sets of probabilities (credal sets) to model the variability in the data-generating distribution. Such credal sets, we argue, may be inferred from a finite sample of training sets. Bounds are derived for the case of finite hypotheses spaces (both assuming realizability or not), as well as infinite model spaces, which directly generalize classical results.

## 1   Introduction

Statistical Learning Theory (SLT) considers the problem of predicting an output $y \in \mathcal{Y}$ given an input $x \in \mathcal{X}$ using a mapping $h : \mathcal{X} \to \mathcal{Y}$, $h \in \mathcal{H}$, called *model* or hypothesis, belonging to a model (or hypotheses) space $\mathcal{H}$. The loss function $l : (\mathcal{X} \times \mathcal{Y}) \times \mathcal{H} \to \mathbb{R}$ measures the error committed by a model $h \in \mathcal{H}$. For instance, the zero-one loss is defined as $l((x, y), h) \doteq \mathbb{I}[y \neq h(x)]$, where $\mathbb{I}$ denotes the indicator function, and assigns a zero value to correct predictions and one to incorrect ones. Input-output pairs are usually assumed to be generated i.i.d. by a probability distribution $P^\star$, which is unknown. The *expected risk* – or *expected loss* – of the model $h$, $L(h) \equiv L_{P^\star}(h) \doteq \mathbb{E}_{P^\star}[l((x, y), h)] = \int_{\mathcal{X} \times \mathcal{Y}} l((x, y), h) P^\star(\mathrm{d}(x, y))$, measures the expected value – taken with respect to $P^\star$ – of loss $l$. The expected risk minimizer $h^\star \in \arg\min_{h \in \mathcal{H}} L(h)$ is any hypothesis in the given model space $\mathcal{H}$ that minimizes the expected risk. Given a training dataset $D = \{(x_1, y_1), \ldots, (x_n, y_n)\}$ whose elements are drawn independently and identically distributed (i.i.d.) from probability distribution $P^\star$, the *empirical risk* of a hypothesis $h$ is the average loss over $D$. The *empirical risk minimizer* (ERM), i.e., the model $\hat{h}$ one actually learns from the training set $D$, is the one minimizing the empirical risk [44]. Statistical Learning Theory seeks upper bounds for the expected risk $L(\hat{h})$ of the ERM $\hat{h}$, and in turn, for the *excess risk*, that is, the difference between $L(\hat{h})$ and the lowest expected risk $L(h^\star)$. This endeavor is pursued under increasingly more relaxed assumptions about the nature of the hypotheses space $\mathcal{H}$. Two common such assumptions are that either the model space is finite, or that there exists a model with zero expected risk (*realizability*).

In real-world situations, however, the data distribution may (and often does) vary, causing issues of *domain adaptation* (DA) [11] or *generalization* (DG) [59]. Domain adaptation and generalization are interrelated yet distinctive concepts in machine learning, as they both deal with the challenges of

transferring knowledge across different domains. The main goal of DA is to adapt a machine learning model trained on source domains to perform well on target domains. In opposition, DG aims to train a model that can generalize well to unseen data/domains not available during training. In simple terms, DA works on the assumption that our source and target domains are related to each other, meaning that they somehow follow a similar data-generating probability distribution. DG, instead, assumes that the trained model should be able to handle unseen target data.

Attempts to derive generalization bounds under more realistic conditions within classical SLT have been made (see Section 2). Those approaches, however, are characterized by a lack of generalizability, and the use of strong assumptions. A more detailed account of the state of the art and their limitations is discussed in Section 2. In opposition to all such proposals, our learning framework leverages *Imprecise Probabilities* (IPs) to provide a radically different solution to the construction of bounds in learning theory.

A hierarchy of formalisms aimed at mathematically modeling the 'epistemic' uncertainty induced by sources such as lack of data, missing data or data which is imprecise in nature [24, 62, 63], IPs have been successfully employed in the design of neural networks providing both better accuracy and uncertainty quantification to predictions [17, 47–49, 64, 73]. To date, however, they have never been considered as a tool to address the foundational issues of statistical learning theory associated with data drifting.

**Contributions**. This paper provides two innovative contributions: (1) the formal definition of a new learning setting in which models are inferred from a (finite) sample of training sets (via either objectivist or subjectivist modeling techniques, as explained in Section 3), rather than a single training set, each assumed to have been generated by a single data distribution (as in classical SLT); (2) the derivation of generalization bounds to the expected risk of a model learned in this new learning setting, under the assumption that the epistemic uncertainty induced by the available training sets can be described by a *credal set* [41], i.e., a convex set of (data generating) probability distributions.

The overall framework is illustrated in Figure 1. Generalized upper bounds under credal uncertainty are derived under three increasingly realistic sets of assumptions, mirroring classical statistical learning theory treatment: (i) finite hypotheses spaces with realizability, (ii) finite hypotheses spaces without realizability, and (iii) infinite hypotheses spaces. We show that the corresponding classical results in SLT are special cases of the ones derived in the present paper.

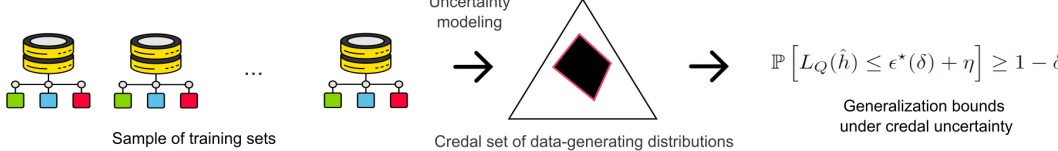

Figure 1: Graphical representation of the proposed learning framework. Given an available finite sample of training sets, each assumed to be generated by a single data distribution, one can learn a credal set $\mathcal{P}$ of data distributions in either a frequentist or subjectivist fashion (Section 3). This allows us to derive generalization bounds under credal uncertainty (Section 4).

**Paper outline**. The paper is structured as follows. First (Section 2) we present the existing work addressing data distribution shifts in learning theory. We then introduce our new learning framework (Section 3). In Section 4 we illustrate the bounds derived under credal uncertainty and show how classical results can be recovered as special cases. Section 5 concludes and outlines future undertakings. We prove our results in Appendix A, and we provide synthetic experiments on our first two main results (Theorems 4.1 and 4.5) in Appendix B.

## 2   Related Work

The standard statistical approach to generalization is based on the assumption that test and training data are i.i.d. according to an unknown distribution. This assumption can fall short in real-world applications: as a result, many recent papers have been focusing on the "Out of Distribution" (OOD) generalization problem, also known as domain generalization (DG), to address the discrepancy between test and training distribution(s) [31, 40, 68]. Extensive surveys of existing methods and

approaches to DG can be found in Wang et al. [71], Zhou et al. [79]. Although several proposals for learning bounds with theoretical guarantees have been made within DA, only a few attempts have been made in the field of DG [28, 60]. Most theoretical attempts have focused on kernel methods, starting from the seminal work of Blanchard et al. [12], spanning to a body of later work (see for example Deshmukh et al. [28], Hu et al. [33], Muandet et al. [57]). In this line of work, assumptions related to boundedness of kernels and continuity of feature maps and loss function render the approaches not directly applicable to broader scenarios.

Other work has focused on providing theoretical grounds using domain-adversarial learning method; in this approach, the authors use a convex combination of source domains in order to approximate the target distribution leveraging H-divergence [2]. Ye et al. [75] have attempted to relax assumptions to provide more general bounds, focusing on feature distribution characteristics; the authors have introduced terms related to stability and discriminative power to calculate the error bound on unseen domains, through the use of an expansion function. Nonetheless, as the authors acknowledge, practical challenges arise concerning the estimation of the expansion function and the choice of a constraint on the top model to improve convergence.

Researchers have also focused on adaptation to new domains over time, treating DG as an online game and the model as a player minimizing the risk associated with introducing new distributions by an adversary at each step [61]. However, in scenarios where the training distribution is significantly outside the convex hull of training distributions [2], or because of unmet strong convexity loss function assumption [61], they fall short from achieving robust generalization. Causality principles have been leveraged in this sense, for example by Bellot and Bareinboim [10], Sheth et al. [67], to provide distributional robustness guarantees using causal diagrams and source domain data. However, causal approaches for improving model robustness across varying domains pose important challenges including reliance on domain knowledge. Researchers have also explored generalization bounds for DG based on the Rademacher complexity, allowing for the approach to be applicable to a broader range of models [42]. Though this simplification has a number of practical benefits, models trained under covariate shift assumptions might suffer in terms of robustness to other distribution shift types. On the empirical analysis side, Gulrajani and Lopez-Paz [31] have provided a comprehensive review of the state of the art. Though a simple ERM was found to outperform other more sophisticated methods in benchmark experiments [21], this approach has been criticized for its non-generalizability. In this direction, Izmailov et al. [35] have highlighted the importance of searching for flat minima in the training process for improved generalization.

All the aforementioned approaches take a point estimate-like, stance (i.e., assuming a single training set) to the derivation of generalization bounds. In this paper, in opposition, we explicitly acknowledge the uncertainty inherent to domain variation in the form of a sample of training sets, each assumed to be generated by a different distribution, and propose a robust and flexible approach representing the resulting epistemic uncertainty via credal sets. Related works on the computational complexity specific to the use of credal sets are discussed in Appendix E.

## 3 Credal Learning

Let us formalize the notion of learning a model from a collection of (training) sets of data, each issued from a different 'domain' characterized by a single, albeit unknown, data-generating probability distribution. Assume that we wish to learn a mapping $h : \mathcal{X} \to \mathcal{Y}$ between an input space $\mathcal{X}$ and an output space $\mathcal{Y}$ (where, once again, the mapping $h$ belongs to a hypotheses space $\mathcal{H}$), having as evidence a finite sample of training sets, $D_1, \ldots, D_N$, $D_i = \{(x_{i,1}, y_{i,1}), \ldots, (x_{i,n_i}, y_{i,n_i})\}$. Assume also that the data in each set $D_i$ has been generated by a distinct probability distribution $P_i^\star$. The question we want to answer is: What sort of guarantees can be derived on the expected risk of a model learned from such a sample of training sets? How do they relate to classical Probably Approximately Correct (PAC) bounds from statistical learning theory?

### 3.1 Objectivist Modeling

While in classical statistical learning theory results are derived assuming no knowledge about the data-generating process, the theorems and corollaries in this paper do require some knowledge, although incomplete, of the true distribution. To be more specific, we will posit that, by leveraging the available evidence $D_1, \ldots, D_N$, the agent is able to elicit a credal set – i.e., a closed and convex

set of probabilities – that contains the true data generating process $P^{\text{true}} \equiv P^{\star}_{N+1}$ for a *new* set of data $D_{N+1}$ (that we call the *test set*), possibly different from $D_1, \ldots, D_N$. As we shall see in Section 4, though, this extra modeling effort allows us to derive stronger results.

There are at least two ways in which such a credal set can be derived, that is, via either an *objectivist* or a *subjectivist* modeling stance. In this section, we present the former. We start by inspecting the frequentist approach to objectivist modeling, considering in particular epsilon-contamination models (Section 3.1.1) and belief functions models (Sections 3.1.2, 3.1.3). A further objectivist model based on fiducial inference [3, 32] is outlined in Appendix D.

### 3.1.1 Epsilon-contamination models

In classical frequentist statistics, given the available dataset, the agent assumes the analytical form of a likelihood $\mathcal{L}$ (not to be confused with the expected loss function, which we denote by a Roman letter $L$), e.g., a Normal or a Gamma distribution. As shown by Huber and Ronchetti [34], though, small perturbations of the specified likelihood can induce substantial differences in the conclusions drawn from the data. A *robust frequentist* agent is thus interested in statistical methods that may not be fully optimal under the ideal 'true' likelihood model, but still lead to reasonable conclusions if the ideal model is only approximately true [8].

To account for this, the agent specifies the class of $\epsilon$-*contaminated* distributions

$$\mathcal{P} = \{P : P = (1 - \epsilon)\mathcal{L} + \epsilon Q, \forall Q\},$$

where $\epsilon$ is some positive quantity in $(0, 1)$, and $Q$ is any distribution on $\mathcal{X} \times \mathcal{Y}$. Wasserman and Kadane [74] show that $\mathcal{P}$ is indeed a (nonempty) credal set. In view of this robust frequentist goal, then, requiring that the true data generating process belongs to $\mathcal{P}$ is a natural assumption.

In our framework, in which a finite sample of $N$ training sets $\{D_i\}_{i=1}^N$ is available, one approach to building the desired credal set is to specify $N$ many likelihoods $\{\mathcal{L}_i\}_{i=1}^N$ and $\epsilon_i$-contaminate each of them to obtain $\mathcal{L}_i = \{P : P = (1 - \epsilon_i)\mathcal{L}_i + \epsilon_i Q, \forall Q\}, i \in \{1, \ldots, N\}$. The credal set $\mathcal{P}$ can then be derived by setting

$$\mathcal{P} = \text{Conv}(\cup_{i=1}^N \mathscr{L}_i),$$

where $\text{Conv}(\cdot)$ denotes the convex hull operator.[1] An immediate consequence of Wasserman and Kadane [74] and references therein is that $\mathcal{P} = \text{Conv}(\cup_{i=1}^N \mathscr{L}_i) = \{P : P(A) \geq \underline{\mathcal{L}}(A), \forall A \subseteq \mathcal{X} \times \mathcal{Y}\}$, where $\underline{\mathcal{L}}(A) = \min_{i \in \{1, \ldots, N\}}(1 - \epsilon_i)\mathcal{L}_i(A)$, for all $A \subset \mathcal{X} \times \mathcal{Y}$. A simple numerical example for such a procedure is given in Appendix C.

### 3.1.2 Belief functions as lower probabilities

An alternative way to derive a credal set from the sample training evidence can be formulated within the framework of the Dempster-Shafer theory of evidence [27, 66].

A *random set* [39, 52, 55, 58] is a set-valued random variable, modeling random experiments in which observations come in the form of sets. In the case of finite sample spaces, they are called *belief functions* [66]. While classical discrete mass functions assign normalized, non-negative values to the *elements* $\omega \in \Omega$ of their sample space, a belief function independently assigns normalized, non-negative mass values to *subsets* of the sample space: $m(A) \geq 0$, for all $A \subseteq \Omega$, $\sum_{A \subseteq \Omega} m(A) = 1$. The belief function associated with a mass function $m$ then measures the total mass of the subsets of each event $A$, $\text{Bel}(A) = \sum_{B \subseteq A} m(B)$.

Crucially, a belief function can be seen as the lower probability (or lower envelope) of the credal set

$$\mathcal{M}(\text{Bel}) = \{P : \Omega \to [0, 1] : \text{Bel}(A) \leq P(A), \forall A \subseteq \Omega\},$$

where $P$ is a data distribution. The dual upper probability to Bel is $\text{Pl}(A) \doteq 1 - \text{Bel}(A^c)$, for all $A \subseteq \Omega$. When restricted to singleton elements, it is called the *contour function*, $\text{pl}(\omega) = \text{Pl}(\{\omega\})$.

---

[1] It is easy to see that the set $\mathcal{P}$ built this way is indeed a credal set. This is because it is (i) convex by definition, and (ii) closed because it is the union of finitely many closed sets.

### 3.1.3 Inferring belief functions from data

There are various ways one can infer a belief (or, equivalently, a plausibility) function from (partial) data, such as a sample of training sets. If a classical likelihood $\mathcal{L}$ having probability density or mass function (pdf/pmf) $\ell$ is available (as assumed in the frequentist paradigm),[2] one can build a belief function by using the normalized likelihood as its contour function. That is, $\mathrm{pl}(\omega) \doteq \frac{\ell(\omega)}{sup_{\omega' \in \Omega} \ell(\omega')}$, for all $\omega \in \Omega$, where $\Omega = \mathcal{X} \times \mathcal{Y}$ is the space where the training pairs live.

As before, in our framework in which a finite sample of $N$ training sets $\{D_i\}_{i=1}^N$ is available, we can specify $N$ many likelihoods $\{\mathcal{L}_i\}_{i=1}^N$, and their corresponding pdf/pmf's $\{\ell_i\}_{i=1}^N$. Then, we can compute $\bar{\ell}(\omega) = \max_{i \in \{1,\dots,N\}} \ell_i(\omega)$, for all $\omega \in \Omega$, and in turn

$$\mathrm{pl}(\omega) = \bar{\ell}(\omega)/sup_{\omega' \in \Omega} \bar{\ell}(\omega'), \tag{1}$$

for all $\omega \in \Omega$.[3] In turn, our credal set is derived as $\mathcal{P} = \{P : \mathrm{d}P/\mathrm{d}\nu = p \leq \mathrm{pl}\}$, where $\mathrm{d}P/\mathrm{d}\nu = p$ is the pdf/pmf associated with distribution $P$ via its Radon-Nikodym derivative with respect to a sigma-finite dominating measure $\nu$. Such construction means that $\mathcal{P}$ includes all distributions whose pdf/pmf's are element-wise dominated by plausibility contour pl.

**Numerical example.** Let $\Omega = \{\omega_1, \omega_2, \omega_3\}$, where $\omega_j = (x_j, y_j)$, $j \in \{1, 2, 3\}$. Suppose also that we observed four sample training sets $D_1, \dots, D_4$ and that we specified the likelihood pmf's $\ell_1, \dots, \ell_4$ as in Table 1.[4] There, we see e.g. how pmf $\ell_1$ assigns a probability of 0.3 to the element $\omega_1$ of the state space $\omega$, and similarly for the other pmf's and the other elements of the state space. It is immediate to see that $\bar{\ell} = (0.4, 0.8, 0.6)^\top$.[5] Then, by Equation (1), we have that $\mathrm{pl} = (0.5, 1, 0.75)^\top$.

We can then derive the lower $\underline{P}$ and upper $\overline{P}$ probabilities of $\mathcal{P} = \{P : \mathrm{d}P/\mathrm{d}\nu = p \leq \mathrm{pl}\}$ on $2^\Omega$ as in Table 2, using the results in Augustin et al. [8, Section 4.4]. That is, $\underline{P}(A) = \max\left\{\sum_{\omega \in A} \underline{P}(\omega), 1 - \sum_{\omega \in A^c} \overline{P}(\omega)\right\}$ and $\overline{P}(A) = \min\left\{\sum_{\omega \in A} \overline{P}(\omega), 1 - \sum_{\omega \in A^c} \underline{P}(\omega)\right\}$. As we can see from the visual representation of $\mathcal{P}$ (the yellow convex region in Figure 2), the probability bounds imposed by the credal set are not too stringent, and in line with the evidence encapsulated in $\ell_1, \dots, \ell_4$. Hence, the assumption that $P^{\text{true}} \equiv P_5^\star \in \mathcal{P}$ is quite plausible.

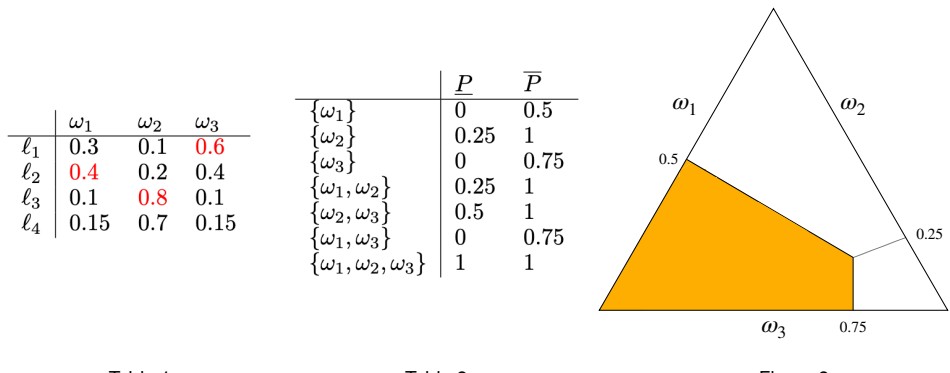

|       | $\omega_1$ | $\omega_2$ | $\omega_3$ |
|-------|------|------|------|
| $\ell_1$ | 0.3  | 0.1  | 0.6  |
| $\ell_2$ | 0.4  | 0.2  | 0.4  |
| $\ell_3$ | 0.1  | 0.8  | 0.1  |
| $\ell_4$ | 0.15 | 0.7  | 0.15 |

Table 1.

|                              | $\underline{P}$ | $\overline{P}$ |
|------------------------------|------|------|
| $\{\omega_1\}$               | 0    | 0.5  |
| $\{\omega_2\}$               | 0.25 | 1    |
| $\{\omega_3\}$               | 0    | 0.75 |
| $\{\omega_1, \omega_2\}$     | 0.25 | 1    |
| $\{\omega_2, \omega_3\}$     | 0.5  | 1    |
| $\{\omega_1, \omega_3\}$     | 0    | 0.75 |
| $\{\omega_1, \omega_2, \omega_3\}$ | 1 | 1  |

Table 2.

Figure 2.

### 3.2 Subjectivist Modeling

Another way of specifying a credal set is by taking a *personalistic* (or subjectivist) route [14, 70]. In this approach, let $\{A_\mathcal{S}\}$ be a finite collection of subsets of $\Omega = \mathcal{X} \times \mathcal{Y}$. The agent first specifies the lower probability $\underline{P}_\mathcal{S}$ on the power set $2^\mathcal{S}$, where $\mathcal{S} = \cup A_\mathcal{S}$ – i.e., the smallest value that the probability of any subset of $\mathcal{S}$ can take on. This can be done, for example, as a result of the empirical distribution, as described below.

---

[2]Here, pdf/pmf $\ell$ is the Radon-Nikodym derivative of $\mathcal{L}$ with respect to a sigma-finite dominating measure $\nu$.

[3]It is easy to see that pl is a well-defined plausibility contour function.

[4]In this case, $\nu$ is the counting measure.

[5]We write the upper likelihood $\bar{\ell}$ in vector form for notational convenience. $^\top$ denotes the transpose.

In our framework in which a finite sample of $N$ training sets $\{D_i\}_{i=1}^N$ is available, we have that $\{A_{\mathcal{S}}\} = \{D_i\}_{i=1}^N$, and so $\mathcal{S} = \cup_{i=1}^N D_i$. Recall that we originally denoted by $P_i^\star$ the true data generating process for training set $D_i$, $i \in \{1, \ldots, N\}$: the empirical distribution $P_i^{\mathrm{emp}}$ is a (non-parametric) estimation of $P_i^\star$. On the other hand, recall that we denoted by $P^{\mathrm{true}} \equiv P_{N+1}^\star$ the true data generating process for the test set of data $D_{N+1}$.

The lower probability $\underline{P}_{\mathcal{S}}$ is defined as follows. For every element $(x, y)$ in $\mathcal{S} = \cup_{i=1}^N D_i$, we let $\underline{P}_{\mathcal{S}}(\{(x,y)\}) \doteq \min\{P_i^{\mathrm{emp}}(\{(x,y)\}) : P_i^{\mathrm{emp}}(\{(x,y)\}) > 0\}$. Requiring $\underline{P}_{\mathcal{S}}(\{(x,y)\}) = \min_i P_i^{\mathrm{emp}}(\{(x,y)\})$ is not enough because, if the training data sets do not overlap, we would end up having lower probability 0 for some singleton that we observed at training time, and hence we would be neglecting some collected evidence. The lower probability $\underline{P}_{\mathcal{S}}$ of all the other non-singleton elements $B$ of $\mathcal{S}$ is computed according to [8, Equation (4.6a)], that is,

$$\underline{P}_{\mathcal{S}}(B) = \max \left\{ \sum_{(x,y) \in B} \underline{P}_{\mathcal{S}}(\{(x,y)\}), 1 - \sum_{(x,y) \in B^c} \max_i P_i^{\mathrm{emp}}(\{(x,y)\}) \right\}. \qquad (2)$$

**Numerical example.** Suppose for simplicity that $\mathcal{X} = \{x\}$, so that $\Omega = \{x\} \times \mathcal{Y} \simeq \mathcal{Y}$, and let $\mathcal{Y} = \{1, \ldots, 10\}$. Suppose $N = 2$, and let $D_1$ be a collection of three 4's, three 5's, and three 6's. Let also $D_2$ be a collection of six 5's and two 6's. Then, $\mathcal{S} = \{4, 5, 6\}$ and $2^{\mathcal{S}} = \{\emptyset, \{4\}, \{5\}, \{6\}, \{4,5\}, \{5,6\}, \{4,6\}, \{4,5,6\}\}$. In turn, $\underline{P}_{\mathcal{S}}(\{4\}) = 1/3$, $\underline{P}_{\mathcal{S}}(\{5\}) = 1/3$, and $\underline{P}_{\mathcal{S}}(\{6\}) = 1/4$. By (2), this implies that $\underline{P}_{\mathcal{S}}(\{4,5\}) = 2/3$, $\underline{P}_{\mathcal{S}}(\{5,6\}) = 2/3$, and $\underline{P}_{\mathcal{S}}(\{4,6\}) = \max\{1/3 + 1/4, 1 - \max_{i \in \{1,2\}} P_i^{\mathrm{emp}}(\{5\})\} = \max\{7/12, 1 - \max\{1/3, 3/4\} = \max\{7/12, 1 - 3/4\} = 7/12$. Of course, $\underline{P}_{\mathcal{S}}(\emptyset) = 0$ and $\underline{P}_{\mathcal{S}}(\mathcal{S}) = 1$.

### 3.2.1 Walley's Natural Extension

Once a lower probability $\underline{P}_{\mathcal{S}}$ on $2^{\mathcal{S}}$ is inferred, it can be (coherently uniquely) extended to a lower probability $\underline{P}$ on the whole sigma-algebra endowed to $\mathcal{X} \times \mathcal{Y}$ through an operator called *natural extension* [69], [70, Sections 3.1.7-3.1.9]. The resulting extended lower probability is such that $\underline{P}(B) = \underline{P}_{\mathcal{S}}(B)$, for all $B \in 2^{\mathcal{S}}$, and a lower probability value $\underline{P}(A)$ is assigned to all the other subsets $A$ of $\mathcal{X} \times \mathcal{Y}$ that are not in $\mathcal{S}$. It is also *coherent* – in Walley's terminology – because, in the behavioral interpretation of probability derived from de Finetti [25, 26], its values cannot be used to construct a bet that would make the agent lose for sure, no matter the outcome of the bet itself.

Once $\underline{P}$ is obtained, the agent can consider the *core* of $\underline{P}$, $\mathcal{M}(\underline{P}) \doteq \{P : \underline{P}(A) \leq P(A), \forall A \subseteq \mathcal{X} \times \mathcal{Y}\}$, i.e., the collection of all the (countably additive) probabilities that set-wise dominate $\underline{P}$. Scholars [20, 50] have shown that $\mathcal{P} = \mathcal{M}(\underline{P})$ is indeed a (nonempty) credal set.

### 3.2.2 Properties of the Core

As shown in Amarante and Maccheroni [5, Example 1] and Amarante et al. [6, Examples 6, 7, 8], given a generic credal set $\mathcal{Q}$ whose lower envelope is $\underline{Q}$ – i.e., a credal set $\mathcal{Q}$ for which $\underline{Q}(\cdot) = \inf_{Q \in \mathcal{Q}} Q(\cdot)$ – we have that $\mathcal{M}(\underline{Q}) \supseteq \mathcal{Q}$. From an information-theoretic perspective, this means that the uncertainty encapsulated in the core of a lower probability $\underline{Q}$ is larger than that in any credal set whose lower envelope is $\underline{Q}$ [13, 15, 16, 18, 30]. In turn, $\mathcal{M}(\underline{Q})$ is the largest credal set the agent can build which represents their partial knowledge. In our learning framework, given the available evidence $D_1, \ldots, D_N$ that the agent uses to derive $\underline{P}_{\mathcal{S}}$, if the agent is confident that $P^{\mathrm{true}}(B) \geq \underline{P}_{\mathcal{S}}(B)$, for all $B \in 2^{\mathcal{S}}$, then it is natural to assume $P^{\mathrm{true}} \equiv P_{N+1}^\star \in \mathcal{M}(\underline{P})$.

## 4  Generalization Bounds under Credal Uncertainty

Consider a credal set $\mathcal{P}$ on $\mathcal{X} \times \mathcal{Y}$ derived as in Section 3, and assume that we collect new evidence in the form of a test set of data $D_{N+1} = \{(x_{N+1,1}, y_{N+1,1}), \ldots, (x_{N+1,n_{N+1}}, y_{N+1,n_{N+1}})\}$. To ease notation, in the following we refer to the newly acquired evidence as $(x_1, y_1), \ldots, (x_n, y_n)$.

An assumption that is common to all the results we present in this section is that $(x_1, y_1), \ldots, (x_n, y_n) \sim P \equiv P^{\mathrm{true}} \equiv P_{N+1}^\star$ i.i.d., and $P \in \mathcal{P}$. This means that either the new evidence comes from one of the distributions that generated $D_1, \ldots, D_N$, or that it is at least *compatible* with the credal set we built from past evidence [30]. That is, either $P$ set-wise dominates the

lower probability $\underline{P}$ of $\mathcal{P}$ (as in Sections 3.1, 3.1.2, and 3.2), or it is set-wise dominated by the upper probability $\overline{P}$ of $\mathcal{P}$ (induced, e.g., by a contour function as in Section 3.1.3). This is a rather natural assumption, especially when the stream of training sets we collect pertains to similar experiments or tasks. For example, this is the case in Continual Learning (CL), where it is customary to assume *task similarity* [47], that is, to posit that the oracle distributions pertaining to all the tasks of interest are all contained in a TV-ball of radius chosen by the user. A more complete discussion on the relation between our credal approach and CL can be found in Appendix F. It is also the case in the healthcare setting, where experts' opinions can be incorporated alongside empirical data (plausible probability distributions) to represent the probability uncertainty, for example, for the prognosis of a disease given a set of patient characteristics/biomarkers [65]. To make sure that the credal set constructed encapsulates most of the "potential distributions needed", a number of approaches can be taken including incremental learning; in this approach the AI model learns and updates knowledge incrementally. As a result, the credal set can be continuously updated (via incremental learning) as new data become available. In this direction, learning health systems are being implemented in practice. These are health systems "in which internal data and experience are systematically integrated with external evidence, and that knowledge is put into practice".

We note, in passing, that assuming $P \in \mathcal{P}$ is less stringent than what is typically done in frequentist statistics, where the data-generating process is assumed to be perfectly captured by the likelihood. We instead posit that the true data generating process for the new evidence available belongs to a credal set $\mathcal{P}$, that was derived by the sample of training sets $D_1, \ldots, D_N$.

Formal ways of checking whether the assumption $P \in \mathcal{P}$ holds exist, e.g., by following what Cella and Martin [19, Section 7] and Javanmardi et al. [36] do for credal-set-based conformal prediction methods, or more general approaches [1, 22, 29, 46, 56].[6] That being said, deriving PAC-like guarantees on the correct distribution $P$ being an element of the credal set $\mathcal{P}$ is a desirable objective - a task that we defer to future work. Here, we focus on formally deriving what the consequences are in terms of generalization bounds.

## 4.1 Realizability and Finite Hypotheses Space

**Theorem 4.1.** *Let* $(x_1, y_1), \ldots, (x_n, y_n) \sim P$ *i.i.d., where* $P$ *is any element of the credal set* $\mathcal{P}$. *Let the empirical risk minimizer be*

$$\hat{h} \in \arg\min_{h \in \mathcal{H}} \frac{1}{n} \sum_{i=1}^{n} l((x_i, y_i), h). \tag{3}$$

*Assume that there exists a realizable hypothesis, that is,* $h^\star \in \mathcal{H}$ *such that* $L_P(h^\star) = 0$, *and that the model space* $\mathcal{H}$ *is finite. Let* $l$ *denote the zero-one loss, and fix any* $\delta \in (0, 1)$. *Then,* $\mathbb{P}[L_P(\hat{h}) \leq \epsilon^\star(\delta)] \geq 1 - \delta$, *where* $\epsilon^\star(\delta)$ *is a well-defined quantity that depends only on* $\delta$ *and on extreme elements* $ex\mathcal{P}$ *of* $\mathcal{P}$, *i.e., those that cannot be written as a convex combination of one another.*

Under the assumptions of finiteness and realizability, Theorem 4.1 gives us a tight probabilistic bound for the expected risk $L_P(\hat{h})$ of the empirical risk minimizer $\hat{h}$. The bound holds for any possible distribution $P$ in the credal set $\mathcal{P}$ that generated the stream of training data. A slightly looser bound depending on the diameter of credal set $\mathcal{P}$ holds if we calculate $L_Q(\hat{h})$ in place of $L_P(\hat{h})$.

**Corollary 4.2.** *Retain the assumptions of Theorem 4.1. Denote by* $Q \in \mathcal{P}$, $Q \neq P$, *a generic distribution in* $\mathcal{P}$ *different from* $P$. *Let* $\Delta_{\mathcal{X} \times \mathcal{Y}}$ *denote the space of all distributions over* $\mathcal{X} \times \mathcal{Y}$, *and endow it with the total variation metric* $d_{TV}$. *Then, pick any* $\eta \in \mathbb{R}_{>0}$. *If the diameter of* $\mathcal{P}$, *denoted by* $diam_{TV}(\mathcal{P})$, *is equal to* $\eta$, *we have that*

$$\mathbb{P}[L_Q(\hat{h}) \leq \epsilon^\star(\delta) + \eta] \geq 1 - \delta,$$

*where* $\epsilon^\star(\delta)$ *is the same quantity as in Theorem 4.1.*

Corollary 4.2 gives us a probabilistic bound for the expected risk $L_Q(\hat{h})$ of the empirical risk minimizer $\hat{h}$, calculated with respect to a "wrong" distribution $Q$ – that is, any distribution in $\mathcal{P}$

---

[6]If we want to avoid to formally check the assumption that $P \in \mathcal{P}$, we need to show on a case-by-case basis that either the credal set covers a non-negligible portion of the distribution class of interest, or that even a small credal set is "good enough" for the analysis at hand.

different from the one generating the new test set of data $D_{N+1}$. We can also give a looser – but easier to compute – bound for $L_P(\hat{h})$.

**Corollary 4.3.** *Under the assumptions of Theorem 4.1, $\epsilon^\star(\delta) \leq \epsilon_{UB}(\delta) \doteq 1/n[\log |\mathcal{H}| + \log(\frac{1}{\delta})]$ and*

$$\mathbb{P}[L_P(\hat{h}) \leq \epsilon_{UB}(\delta)] \geq 1 - \delta, \quad \forall P \in \Delta_{\mathcal{X} \times \mathcal{Y}}.$$

Notice how $\epsilon_{\text{UB}}(\delta)$ is a uniform bound, that is, a bound that holds for all possible distributions on $\mathcal{X} \times \mathcal{Y}$, not just those in $\mathcal{P}$. Strictly speaking, this means that we do not need to come up with a credal set $\mathcal{P}$ to find such a bound. Observe, though, that the bound $\epsilon^\star(\delta)$ in Theorem 4.1 is tighter, as it leverages the training evidence encoded in the credal set $\mathcal{P}$. A synthetic experiment confirming this, and studying other interesting properties of $\epsilon^\star(\delta)$ and $\epsilon_{\text{UB}}(\delta)$, can be found in Appendix B.

By the proof of Theorem 4.1, we have that $L_P(\hat{h})$ behaves as $\mathcal{O}(\log |\cup_{P^{\text{ex}} \in \text{ex}\mathcal{P}} B_{P^{\text{ex}}}|/n)$, which is faster than the rate $\mathcal{O}(\log |\mathcal{H}|/n)$ that we find in Corollary 4.3, since $\cup_{P^{\text{ex}} \in \text{ex}\mathcal{P}} B_{P^{\text{ex}}} \subseteq \mathcal{H}$.[7] Roughly, $B_{P^{\text{ex}}}$ is the set of "bad hypotheses" according to $P^{\text{ex}} \in \text{ex}\mathcal{P}$. That is, those $h$'s for which $L_{P^{\text{ex}}}(h)$ is larger than 0. A formal definition is given in the proof of Theorem 4.1. The modeling effort required by producing credal set $\mathcal{P}$ is therefore rewarded with a tighter bound and a faster rate.

Notice that Corollary 4.3 corresponds to Liang [44, Theorem 4]: we obtain a classical result as a special case of our more general theorem.

Let us now allow for distribution drift in the new test set of data $D_{N+1}$.

**Corollary 4.4.** *Consider a natural number $k < n$. Let $(x_1, y_1), \ldots, (x_k, y_k) \sim P_1$ i.i.d., and $(x_{k+1}, y_{k+1}), \ldots, (x_n, y_n) \sim P_2$ i.i.d., where $P_1, P_2$ are two generic elements of credal set $\mathcal{P}$. Retain the other assumptions of Theorem 4.1. Then,*

$$\mathbb{P}\left[L_{P_1}(\hat{h}_1) + L_{P_2}(\hat{h}_2) \leq \epsilon^\star(\delta) \frac{n^2}{k(n-k)}\right] \geq 1 - \delta, \tag{4}$$

*where $\epsilon^\star(\delta)$ is the same quantity as in Theorem 4.1, and*

$$\hat{h}_1 \in \arg\min_{h \in \mathcal{H}} \left\{\frac{1}{k} \sum_{i=1}^k l((x_i, y_i), h)\right\}, \quad \hat{h}_2 \in \arg\min_{h \in \mathcal{H}} \left\{\frac{1}{n-k} \sum_{i=k+1}^n l((x_i, y_i), h)\right\}.$$

Corollary 4.4 gives us a bound similar to the one in Theorem 4.1 when distribution drift is allowed. The price we pay for it is that it is looser. As a result of Corollary 4.3, for a looser but easier to compute bound, we can substitute $\epsilon^\star(\delta)$ with $\epsilon_{\text{UB}}(\delta)$.

## 4.2 No Realizability and Finite Hypotheses Space

Let us now relax the realizability assumption in Theorem 4.1.

**Theorem 4.5.** *Let $(x_1, y_1), \ldots, (x_n, y_n) \sim P$ i.i.d., where $P$ is any element of the credal set $\mathcal{P}$. Assume that the model space $\mathcal{H}$ is finite. Let $l$ be the zero-one loss, $\hat{h}$ the empirical risk minimizer, and $h^\star$ the best theoretical model. Fix any $\delta \in (0, 1)$. Then, $\mathbb{P}[L_P(\hat{h}) - L_P(h^\star) \leq \epsilon^{\star\star}(\delta)] \geq 1 - \delta$, where $\epsilon^{\star\star}(\delta)$ is a well-defined quantity that depends only on $\delta$ and on the elements of $\text{ex}\mathcal{P}$.*

As we did in Section 4.1, we can also show that the "wrong" expected risk $L_Q(\hat{h})$ – that is, the expected risk computed according to $Q \in \mathcal{P}$ different from the one generating the new evidence $D_{N+1}$ – concentrates around the expected risk $L_P(h^\star)$ evaluated at the best theoretical model $h^\star$.

**Corollary 4.6.** *Retain the assumptions of Theorem 4.5. Denote by $Q \in \mathcal{P}$, $Q \neq P$, a generic distribution in $\mathcal{P}$ different from $P$. Pick any $\eta \in \mathbb{R}_{>0}$; if $diam_{TV}(\mathcal{P}) = \eta$, we have that*

$$\mathbb{P}[L_Q(\hat{h}) - L_P(h^\star) \leq \epsilon^{\star\star}(\delta) + \eta] \geq 1 - \delta,$$

*where $\epsilon^{\star\star}(\delta)$ is the same quantity as in Theorem 4.5.*

Similarly to Corollary 4.3, we can give a looser – but easier to compute – bound for $L_P(\hat{h}) - L_P(h^\star)$.

---

[7]Notice how, if $\mathcal{P}$ has finitely many extreme elements – which happens, e.g., if we put $\mathcal{P} = \text{Conv}(\{\mathcal{L}_i\}_{i=1}^N)$, another frequentist way of deriving a credal set – then $\cup_{P^{\text{ex}} \in \text{ex}\mathcal{P}} B_{P^{\text{ex}}}$ is a finite union, hence easier to compute.

**Corollary 4.7.** *Retain the assumptions of Theorem 4.5. Then, $\epsilon^{\star\star}(\delta) \leq \epsilon'_{UB}(\delta) \doteq \sqrt{\frac{2(\log|\mathcal{H}|+\log(\frac{2}{\delta}))}{n}}$. In turn, $\mathbb{P}[L_P(\hat{h}) - L_P(h^\star) \leq \epsilon'_{UB}(\delta)] \geq 1 - \delta$, for all $P \in \Delta_{\mathcal{X} \times \mathcal{Y}}$.*

The main difference with respect to Theorem 4.1 is that in Theorem 4.5, $L_P(\hat{h}) - L_P(h^\star)$ behaves as $\mathcal{O}(\sqrt{\log|B'_{\text{ex}\mathcal{P}}|/n})$, which is slower than what we had in Theorem 4.1. This is due to the relaxation of the realizability hypothesis. Just like before, though, we have that $\mathcal{O}(\sqrt{\log|B'_{\text{ex}\mathcal{P}}|/n})$ is faster than the rate $\mathcal{O}(\sqrt{\log|\mathcal{H}|/n})$ that we find in Corollary 4.7. This is because $B'_{\text{ex}\mathcal{P}} \subseteq \mathcal{H}$.

Roughly, $B'_{\text{ex}\mathcal{P}}$ is the set of "bad hypotheses" according to at least one $P^{\text{ex}} \in \text{ex}\mathcal{P}$. That is, those $h$'s for which $|\hat{L}(h) - L_{P^{\text{ex}}}(h)|$ is larger than 0, for at least one $P^{\text{ex}}$. A formal definition is given in the proof of Theorem 4.5. Notice that Corollary 4.7 corresponds to Liang [44, Theorem 7]: we obtain a classical result as a special case of our more general theorem.

Let us now allow for distribution drift. To improve notation clarity, in the following we let $h_P^\star$ denote an element of $\arg\min_{h \in \mathcal{H}} L_P(h)$, for a distribution $P \in \mathcal{P}$.

**Corollary 4.8.** *Consider a natural number $k < n$. Let $(x_1, y_1), \ldots, (x_k, y_k) \sim P_1$ i.i.d., and $(x_{k+1}, y_{k+1}), \ldots, (x_n, y_n) \sim P_2$ i.i.d., where $P_1, P_2$ are two generic elements of credal set $\mathcal{P}$. Retain the other assumptions of Theorem 4.5. Then,*

$$\mathbb{P}\Big[(L_{P_1}(\hat{h}_1) - L_{P_1}(h_{P_1}^\star)) + (L_{P_2}(\hat{h}_2) - L_{P_2}(h_{P_2}^\star)) \leq \epsilon^{\star\star}(\delta)\sqrt{\frac{n}{k(n-k)}}(\sqrt{k} + \sqrt{n-k})\Big] \geq 1 - \delta,$$

*where $\epsilon^{\star\star}(\delta)$ is the same quantity as in Theorem 4.5, and $\hat{h}_1$ and $\hat{h}_2$ are defined as in Corollary 4.4.*

Corollary 4.8 tells us that the excess risk is also bounded in the presence of distribution drift. The price we pay for allowing distribution shift is a looser bound. As a result of Corollary 4.7, for a looser but easier to compute bound, we can substitute $\epsilon^{\star\star}(\delta)$ with $\epsilon'_{UB}(\delta)$.

### 4.3  No Realizability and Infinite Hypotheses Space

We now relax also the finite hypotheses space assumption in Theorem 4.1.

**Theorem 4.9.** *Let $(x_1, y_1), \ldots, (x_n, y_n) \sim P$ i.i.d., where $P$ is any element of credal set $\mathcal{P}$. Let $l$ denote the zero-one loss, $\hat{h}$ the empirical risk minimizer and $h^\star$ the best theoretical model. Fix any $\delta \in (0, 1)$. Then,*

$$\mathbb{P}\left[L_P(\hat{h}) - L_P(h^\star) \leq \epsilon^{\star\star\star}(\delta)\right] \geq 1 - \delta, \tag{5}$$

*for all $P \in \mathcal{P}$. Here, $\epsilon^{\star\star\star}(\delta) \doteq 4\overline{R}_{n,P^{\text{ex}}}(\mathcal{A}) + \sqrt{\frac{2\log(2/\delta)}{n}}$, where $\overline{R}_{n,P^{\text{ex}}}(\mathcal{A}) \doteq \sup_{P^{\text{ex}} \in \text{ex}\mathcal{P}} R_{n,P^{\text{ex}}}(\mathcal{A})$ and*

$$R_{n,P^{\text{ex}}}(\mathcal{A}) \doteq \mathbb{E}_{P^{\text{ex}}}\left[\sup_{h \in \mathcal{H}} \frac{1}{n} \sum_{i=1}^{n} \sigma_i l((x_i, y_i), h)\right]. \tag{6}$$

*In (6), $\sigma_1, \ldots, \sigma_n \sim \text{Unif}(\{-1, 1\})$, and $\mathcal{A} \doteq \{(x, y) \mapsto l((x, y), h) : h \in \mathcal{H}\}$.*

$R_{n,P^{\text{ex}}}(\mathcal{A})$ is a slight modification of the classical *Rademacher complexity* of class $\mathcal{A}$,[8] given by

$$R_n(\mathcal{A}) \equiv R_{n,P}(\mathcal{A}) \doteq \mathbb{E}_P\left[\sup_{h \in \mathcal{H}} \frac{1}{n} \sum_{i=1}^{n} \sigma_i l((x_i, y_i), h)\right],$$

where the expectation is taken with respect to the same distribution $P$ from which the data points $(x_1, y_1), \ldots, (x_n, y_n)$ are drawn. We consider $R_{n,P^{\text{ex}}}(\mathcal{A})$ instead of $R_{n,P}(\mathcal{A})$ because, since $\mathcal{P}$ is a credal set, $P$ can be written as a convex combination of the extreme elements of $\mathcal{P}$, and $\sup_{P \in \mathcal{P}} R_{n,P}(\mathcal{A}) = \sup_{P^{\text{ex}} \in \text{ex}\mathcal{P}} R_{n,P^{\text{ex}}}(\mathcal{A})$. If the credal set is *finitely generated*, that is, if it has

---

[8]Class $\mathcal{A}$ is the *loss class*, and it is the composition of the zero-one loss function with each of the hypotheses in $\mathcal{H}$ [44, Page 70]. The Rademacher complexity of $\mathcal{A}$ measures how well the best element of $\mathcal{H}$ fits random noise (coming from the $\sigma_i$'s) [9], [44, Page 69].

finitely many extreme elements (see footnote 7), then it is easier to compute $\epsilon^{\star\star\star}(\delta)$: we only need to compute a maximum in place of a supremum.

As we show in Corollary 4.10, Theorem 4.9 generalizes Liang [44, Theorem 9]. This latter focuses only on the "true" probability $P_{N+1}^\star \equiv P^{\text{true}}$ on $\mathcal{X} \times \mathcal{Y}$, while our result holds for all the plausible distributions in credal set $\mathcal{P}$. This grants us to hedge against distribution misspecification.

Let us pause here to add a clarification. In real-world applications, we effectively cannot compute $R_{n,P^{\text{true}}}(\mathcal{A})$, since the distribution $P^{\text{true}}$ is unknown. While $R_{n,P^{\text{true}}}(\mathcal{A})$ can be approximated via the *empirical Rademacher complexity* $\hat{R}_n(\mathcal{A})$ [44, Equation (219)], whose expected value is indeed $R_{n,P^{\text{true}}}(\mathcal{A})$, doing so has at least two drawbacks: **(1)** When the number of data points $n \equiv n_{D_{N+1}}$ is not "large enough", this may lead to a poor approximation of the classical bound (Equation (10) in Appendix A); **(2)** The test set of data $D_{N+1} = \{(x_i, y_i)\}_{i=1}^n$ may well be a realization from the tail of distribution $P^{\text{true}} \equiv P_{N+1}^\star$. The empirical Rademacher complexity $\hat{R}_n(\mathcal{A})$, then, would be a poor approximation of $R_{n,P^{\text{true}}}(\mathcal{A})$. In opposition, while $\overline{R}_{n,P^{\text{ex}}}(\mathcal{A})$ is more conservative, it can be computed explicitly – since we know the credal set $\mathcal{P}$ and its extreme elements $\text{ex}\mathcal{P}$ – and it leads to a bound that, although looser, holds for all $P \in \mathcal{P}$.

**Corollary 4.10.** *Retain the assumptions of Theorem 4.9. If $\mathcal{P}$ is the singleton $\{P^{\text{true}}\}$ (i.e., all the training datasets $D_1, \ldots, D_N$ are generated by the same distribution as the new test set $D_{N+1}$), we retrieve Liang [44, Theorem 9].*

We then derive a more general version of Corollary 4.6.

**Corollary 4.11.** *Retain the assumptions of Theorem 4.9. Denote by $Q \in \mathcal{P}$, $Q \neq P$, a generic distribution in $\mathcal{P}$ different from $P$. Pick any $\eta \in \mathbb{R}_{>0}$; if $diam_{TV}(\mathcal{P}) = \eta$, we have that*

$$\mathbb{P}[L_Q(\hat{h}) - L_P(h^\star) \leq \epsilon^{\star\star\star}(\delta) + \eta] \geq 1 - \delta,$$

*where $\epsilon^{\star\star\star}(\delta)$ is the same quantity as in Theorem 4.9.*

Finally, we once again allow for distribution drift.

**Corollary 4.12.** *Consider a natural number $k < n$. Let $(x_1, y_1), \ldots, (x_k, y_k) \sim P_1$ i.i.d., and $(x_{k+1}, y_{k+1}), \ldots, (x_n, y_n) \sim P_2$ i.i.d., where $P_1, P_2$ are two generic elements of credal set $\mathcal{P}$. Retain the other assumptions of Theorem 4.9, and let $\epsilon_{shift}^{\star\star\star} \doteq 4[\overline{R}_{k,P^{\text{ex}}}(\mathcal{A}) + \overline{R}_{n-k,P^{\text{ex}}}(\mathcal{A})] + \sqrt{\frac{2\log(2/\delta)}{n(n-k)}}(\sqrt{n-k} + \sqrt{n})$. Then,*

$$\mathbb{P}[(L_{P_1}(\hat{h}_1) - L_{P_1}(h_{P_1}^\star)) + (L_{P_2}(\hat{h}_2) - L_{P_2}(h_{P_2}^\star)) \leq \epsilon_{shift}^{\star\star\star}] \geq 1 - \delta,$$

*where $\hat{h}_1$ and $\hat{h}_2$ are defined as in Corollary 4.4.*

Similar considerations as the ones after Corollaries 4.4 and 4.8 hold in this more general case as well.

## 5 Conclusions

In this paper, we laid the foundations of a more general Statistical Learning Theory (SLT), that we called Credal Learning Theory (CLT). We generalized some of the most important results of classical SLT to allow for drift and misspecification of the data-generating process. We did so by considering sets of probabilities (credal sets), instead of single distributions. The modeling effort needed to elicit credal sets is paid off in terms of the tightness of the resulting bounds.

**Limitations.** (i) We only consider the zero-one loss in our results (we did so to be able to directly build on the classical results in Liang [44, Chapter 3]). (ii) We assume that the true distribution which the elements of the new test set $D_{N+1}$ are sampled from, belongs to the credal set that we derive at training time.

**Future work.** In the future, we plan to further our undertaking, for instance by (i) modeling the epistemic uncertainty induced by domain variation through random sets rather than credal sets, (ii) comparing our method with robust learning [23], (iii) extending our results to different losses, and (iv) deriving PAC-like guarantees on the correct distribution $P$ being an element of the credal set $\mathcal{P}$. We also intend to validate our findings on real datasets.

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

## A Proofs

*Proof of Theorem 4.1.* The proof builds on that of Liang [44, Theorem 4]. Fix any $\epsilon > 0$, and any $P \in \mathcal{P}$. Assume that the training dataset is given by $n$ i.i.d. draws from $P$. We want to bound the probability that $L_P(\hat{h}) > \epsilon$. Define $B_P \doteq \{h \in \mathcal{H} : L_P(h) > \epsilon\}$. It is the set of "bad hypotheses" according to distribution $P$. As a consequence, we can write $\mathbb{P}[L_P(\hat{h}) > \epsilon] = \mathbb{P}[\hat{h} \in B_P]$. Recall that the empirical risk of the empirical risk minimizer is 0, that is, $\hat{L}(\hat{h}) = 0$.[9] So if the empirical risk minimizer is a bad hypothesis according to $P$, that is, if $\hat{h} \in B_P$, then some bad hypothesis (according to $P$) must have zero empirical risk. In turn,

$$\mathbb{P}[\hat{h} \in B_P] \leq \mathbb{P}[\exists h \in B_P : \hat{L}(h) = 0].$$

Let us bound $\mathbb{P}[\hat{L}(h) = 0]$ for a fixed $h \in B_P$. Given our choice of zero-one loss, on each example, hypothesis $h$ does not err with probability $1 - L_P(h)$. Since the training examples are i.i.d. and $L_P(h) > \epsilon$ for all $h \in B_P$, then

$$\mathbb{P}[\hat{L}(h) = 0] = (1 - L_P(h))^n \leq (1 - \epsilon)^n \leq \exp(-\epsilon n). \tag{7}$$

---

[9]Indeed, at least $\hat{L}(h^\star) = L_P(h^\star) = 0$.

Applying the union bound, we obtain

$$
\begin{aligned}
\mathbb{P}[\exists h \in B_P : \hat{L}(h) = 0] &\leq \sum_{h \in B_P} \mathbb{P}[\hat{L}(h) = 0] \\
&\leq |B_P| \exp(-\epsilon n) \\
&\leq |\cup_{P \in \mathcal{P}} B_P| \exp(-\epsilon n) \\
&= |\cup_{P^{\mathrm{ex}} \in \mathrm{ex}\mathcal{P}} B_{P^{\mathrm{ex}}}| \exp(-\epsilon n) \\
&\doteq \delta.
\end{aligned}
$$

The penultimate equality comes from $\mathcal{P}$ being a credal set, by the Bauer Maximum Principle and the linearity of the expectation operator. Rearranging the terms we get

$$
\epsilon^{\star}(\delta) \equiv \epsilon = \frac{\log |\cup_{P^{\mathrm{ex}} \in \mathrm{ex}\mathcal{P}} B_{P^{\mathrm{ex}}}| + \log(1/\delta)}{n}.
$$

In turn, this implies that $\mathbb{P}[L_P(\hat{h}) \leq \epsilon^{\star}(\delta)] \geq 1 - \delta$. $\qquad \square$

*Proof of Corollary 4.2.* Let $A_{\hat{h}} \doteq \{(x, y) \in \mathcal{X} \times \mathcal{Y} : y \neq \hat{h}(x)\} \in \mathcal{A}_{\mathcal{X} \times \mathcal{Y}}$. Notice that

$$
\begin{aligned}
L_P(\hat{h}) &= \mathbb{E}_P[\mathbb{I}(y \neq \hat{h}(x))] = P(A_{\hat{h}}), \\
L_Q(\hat{h}) &= \mathbb{E}_Q[\mathbb{I}(y \neq \hat{h}(x))] = Q(A_{\hat{h}}).
\end{aligned}
$$

Recall that $\mathrm{diam}_{TV}(\mathcal{P}) \doteq \sup_{P,Q \in \mathcal{P}} \sup_{A \in \mathcal{A}_{\mathcal{X} \times \mathcal{Y}}} |P(A) - Q(A)|$. As a consequence, we have that $L_Q(\hat{h}) = L_P(\hat{h}) + \zeta_Q$, where $\zeta_Q$ is a quantity in $[-\eta, \eta]$ depending on $Q$. Given our assumption on the diameter, then, $L_P(\hat{h}) + \zeta_Q \leq L_P(\hat{h}) + \eta$, so $L_Q(\hat{h}) - \eta \leq L_P(\hat{h})$. In turn this implies that

$$
\mathbb{P}\left[L_Q(\hat{h}) - \eta \leq \epsilon^{\star}(\delta)\right] \geq \mathbb{P}\left[L_P(\hat{h}) \leq \epsilon^{\star}(\delta)\right].
$$

The proof is concluded by noting that $\mathbb{P}[L_Q(\hat{h}) - \eta \leq \epsilon^{\star}(\delta)] = \mathbb{P}[L_Q(\hat{h}) \leq \epsilon^{\star}(\delta) + \eta]$, and that $\mathbb{P}[L_P(\hat{h}) \leq \epsilon^{\star}(\delta)] \geq 1 - \delta$ by Theorem 4.1. $\qquad \square$

*Proof of Corollary 4.3.* Since $\cup_{P^{\mathrm{ex}} \in \mathrm{ex}\mathcal{P}} B_{P^{\mathrm{ex}}} \subseteq \mathcal{H}$, it is immediate to see that $\epsilon^{\star}(\delta) \leq \epsilon_{\mathrm{UB}}(\delta)$. In turn,

$$
\mathbb{P}\left[\sup_{P \in \mathcal{P}} L_P(\hat{h}) \leq \epsilon_{\mathrm{UB}}(\delta)\right] \geq 1 - \delta,
$$

or equivalently, $\mathbb{P}[L_P(\hat{h}) \leq \epsilon_{\mathrm{UB}}(\delta)] \geq 1 - \delta$, for all $P \in \mathcal{P}$. $\qquad \square$

*Proof of Corollary 4.4.* From Theorem 4.1, we have that

$$
\mathbb{P}\left[L_{P_1}(\hat{h}_1) \leq \frac{\log |\cup_{P^{\mathrm{ex}} \in \mathrm{ex}\mathcal{P}} B_{P^{\mathrm{ex}}}| + \log(1/\delta)}{k}\right] \geq 1 - \delta,
$$

and that

$$
\mathbb{P}\left[L_{P_2}(\hat{h}_2) \leq \frac{\log |\cup_{P^{\mathrm{ex}} \in \mathrm{ex}\mathcal{P}} B_{P^{\mathrm{ex}}}| + \log(1/\delta)}{n - k}\right] \geq 1 - \delta.
$$

The result, then, is an immediate consequence of the additivity of the expectation operator and of probability $\mathbb{P}$. $\qquad \square$

*Proof of Theorem 4.5.* The proof builds on that of Liang [44, Theorem 7]. Fix any $\epsilon > 0$, and any $P \in \mathcal{P}$. Assume that the training dataset is given by $n$ i.i.d. draws from $P$. By Liang [44, Equations (158) and (186)], we have that

$$
\begin{aligned}
\mathbb{P}[L_P(\hat{h}) - L_P(h^{\star}) > \epsilon] &\leq \mathbb{P}\left[\sup_{h \in \mathcal{H}} \left|\hat{L}(h) - L_P(h)\right| > \frac{\epsilon}{2}\right] \\
&< |\mathcal{H}| \cdot 2 \exp\left(-2n \left(\frac{\epsilon}{2}\right)^2\right) \\
&\doteq \delta(\epsilon).
\end{aligned} \tag{8}
$$

Notice though, that we can improve on this bound, since we know that $P \in \mathcal{P}$, a credal set. Let $B'_P \doteq \{h \in \mathcal{H} : |\hat{L}(h) - L_P(h)| > \epsilon/2\}$ be the set of "bad hypotheses" according to $P$. Then, it is immediate to see that

$$\sup_{h \in \mathcal{H}} \left| \hat{L}(h) - L_P(h) \right| = \sup_{h \in B'_P} \left| \hat{L}(h) - L_P(h) \right|.$$

Notice though that we do not know $P$; we only know it belongs to $\mathcal{P}$. Hence, we need to consider the set $B'_\mathcal{P}$ of bad hypotheses according to all the elements of $\mathcal{P}$, that is, $B'_\mathcal{P} \doteq \{h \in \mathcal{H} : \exists P \in \mathcal{P}, |\hat{L}(h) - L_P(h)| > \epsilon/2\} = \cup_{P \in \mathcal{P}} B'_P$. Since $\mathcal{P}$ is a credal set, by the Bauer Maximum Principle and the linearity of the expectation operator we have that $B'_\mathcal{P} = B'_{\mathrm{ex}\mathcal{P}} \doteq \{h \in \mathcal{H} : \exists P^{\mathrm{ex}} \in \mathrm{ex}\mathcal{P}, |\hat{L}(h) - L_P(h)| > \epsilon/2\} = \cup_{P^{\mathrm{ex}} \in \mathrm{ex}\mathcal{P}} B'_{P^{\mathrm{ex}}}$. Hence, we obtain

$$\sup_{h \in \mathcal{H}} \left| \hat{L}(h) - L_P(h) \right| = \sup_{h \in B'_{\mathrm{ex}\mathcal{P}}} \left| \hat{L}(h) - L_P(h) \right|.$$

In turn, (8) implies that

$$\mathbb{P}\big[ L_P(\hat{h}) - L_P(h^\star) > \epsilon \big] \leq \mathbb{P}\left[ \sup_{h \in B'_{\mathrm{ex}\mathcal{P}}} \left| \hat{L}(h) - L_P(h) \right| > \frac{\epsilon}{2} \right]$$

$$< |B'_{\mathrm{ex}\mathcal{P}}| \cdot 2 \exp\left( -2n \left( \frac{\epsilon}{2} \right)^2 \right)$$

$$\doteq \delta_{\mathrm{ex}\mathcal{P}}.$$

Rearranging, we obtain

$$\epsilon = \sqrt{\frac{2 \left( \log |B'_{\mathrm{ex}\mathcal{P}}| + \log\left( \frac{2}{\delta_{\mathrm{ex}\mathcal{P}}} \right) \right)}{n}}, \tag{9}$$

so if $\delta$ is fixed, we can write $\epsilon \equiv \epsilon^{\star\star}(\delta)$. In turn, this implies that $\mathbb{P}[L_P(\hat{h}) - L_P(h^\star) > \epsilon^{\star\star}(\delta)] < \delta$, or equivalently, $\mathbb{P}[L_P(\hat{h}) - L_P(h^\star) \leq \epsilon^{\star\star}(\delta)] \geq 1 - \delta$. $\qquad\square$

*Proof of Corollary 4.6.* The first part of the proof is very similar to that of Corollary 4.2. Given our assumption on the diameter, we have that $L_Q(\hat{h}) - L_P(h^\star) = L_P(\hat{h}) + \zeta_Q - L_P(h^\star)$, where $\zeta_Q$ is a quantity in $[-\eta, \eta]$ depending on $Q$. Then, $L_P(\hat{h}) + \zeta_Q - L_P(h^\star) \leq L_P(\hat{h}) + \eta - L_P(h^\star)$, so $L_Q(\hat{h}) - \eta - L_P(h^\star) \leq L_P(\hat{h}) - L_P(h^\star)$. In turn this implies that $\mathbb{P}\left[ L_Q(\hat{h}) - \eta - L_P(h^\star) \leq \epsilon^{\star\star}(\delta) \right] \geq \mathbb{P}\left[ L_P(\hat{h}) - L_P(h^\star) \leq \epsilon^{\star\star}(\delta) \right]$. The proof is concluded by noting that $\mathbb{P}[L_Q(\hat{h}) - \eta - L_P(h^\star) \leq \epsilon^{\star\star}(\delta)] = \mathbb{P}[L_Q(\hat{h}) - L_P(h^\star) \leq \epsilon^{\star\star}(\delta) + \eta]$, and that $\mathbb{P}[L_P(\hat{h}) - L_P(h^\star) \leq \epsilon^{\star\star}(\delta)] \geq 1 - \delta$ by Theorem 4.5. $\qquad\square$

*Proof of Corollary 4.7.* Since $\cup_{P^{\mathrm{ex}} \in \mathrm{ex}\mathcal{P}} B'_{P^{\mathrm{ex}}} \subseteq \mathcal{H}$, it is immediate to see that $\epsilon^{\star\star}(\delta) \leq \epsilon'_{\mathrm{UB}}(\delta)$. In turn,

$$\mathbb{P}\left[ \sup_{P \in \mathcal{P}} \left( L_P(\hat{h}) - L_P(h^\star) \right) \leq \epsilon'_{\mathrm{UB}}(\delta) \right] \geq 1 - \delta,$$

or equivalently, $\mathbb{P}[L_P(\hat{h}) - L_P(h^\star) \leq \epsilon'_{\mathrm{UB}}(\delta)] \geq 1 - \delta$, for all $P \in \mathcal{P}$. $\qquad\square$

*Proof of Corollary 4.8.* From Theorem 4.5, we have that

$$\mathbb{P}\left[ L_{P_1}(\hat{h}_1) - L_{P_1}\left( h^\star_{P_1} \right) \leq \sqrt{\frac{2 \left( \log |B'_{\mathrm{ex}\mathcal{P}}| + \log\left( \frac{2}{\delta} \right) \right)}{k}} \right] \geq 1 - \delta,$$

and that

$$\mathbb{P}\left[ L_{P_2}(\hat{h}_2) - L_{P_2}\left( h^\star_{P_2} \right) \leq \sqrt{\frac{2 \left( \log |B'_{\mathrm{ex}\mathcal{P}}| + \log\left( \frac{2}{\delta} \right) \right)}{n - k}} \right] \geq 1 - \delta.$$

The result, then, is an immediate consequence of the additivity of the expectation operator and of probability $\mathbb{P}$. $\qquad\square$

*Proof of Theorem 4.9.* Fix any $\delta \in (0, 1)$. In Liang [44, Theorem 9], the author shows that for a fixed probability measure $P$ on $\mathcal{X} \times \mathcal{Y}$, we have that

$$L_P(\hat{h}) - L_P(h^\star) \leq 4R_{n,P}(\mathcal{A}) + \sqrt{\frac{2\log(2/\delta)}{n}} \tag{10}$$

holds with probability at least $1 - \delta$, where $R_{n,P}$ is defined analogously as in (6). The result in (5), then, follows from $\mathcal{P}$ being a credal set, and the expectation being a linear operator. □

*Proof of Corollary 4.10.* Immediate from Theorem 4.9. □

*Proof of Corollary 4.11.* The proof is very similar to that of Corollary 4.6. □

*Proof of Corollary 4.12.* From Theorem 4.9, we have that

$$\mathbb{P}\left[L_{P_1}(\hat{h}_1) - L_{P_1}(h_1^\star) \leq 4\overline{R}_{k,P^{\mathrm{ex}}}(\mathcal{A}) + \sqrt{\frac{2\log(2/\delta)}{k}}\right] \geq 1 - \delta,$$

and that

$$\mathbb{P}\left[L_{P_2}(\hat{h}_2) - L_{P_2}(h_2^\star) \leq 4\overline{R}_{n-k,P^{\mathrm{ex}}}(\mathcal{A}) + \sqrt{\frac{2\log(2/\delta)}{n-k}}\right] \geq 1 - \delta.$$

The result, then, is an immediate consequence of the additivity of the expectation operator and of probability $\mathbb{P}$. □

## B  Synthetic Experiments on Theorems 4.1 and 4.5

In this section, we perform synthetic experiments to show that the bounds we find in Theorems 4.1 and 4.5 are indeed tighter than the classical SLT ones reported in Corollaries 4.3 and 4.7, respectively. In recent literature, studies by Amit et al. [7], Kacham and Woodruff [38], Li and Liu [43] have conducted synthetic experiments in a similar manner. These works are mainly theoretical in nature, but they also acknowledge the importance of experimental validation with preliminary analysis.

**Experiment 1:** Let the available training sets be $D_1, D_2, D_3$. Assume, for simplicity, that $\Omega = \mathcal{X} \times \mathcal{Y} = \{x\} \times \mathbb{R} \simeq \mathbb{R}$. Suppose that we specified the likelihood pdfs $\ell_1 = \mathcal{N}(-5, 1)$, $\ell_2 = \mathcal{N}(0, 1)$, and $\ell_3 = \mathcal{N}(5, 1)$. Call $\mathcal{L}_1, \mathcal{L}_2, \mathcal{L}_3$ their respective probability measures, and derive the credal set $\mathcal{P}$ as we did in footnote 7. That is, let $\mathcal{P} = \mathrm{Conv}(\{\mathcal{L}_i\}_{i=1}^3)$. We determine the credal set in this way because it is then easy to find its extreme elements $\mathrm{ex}\mathcal{P}$. Indeed, it is immediate to notice that $\mathrm{ex}\mathcal{P} = \{\mathcal{L}_i\}_{i=1}^3$. Let now $D_{N+1} \equiv D_4$ be a collection of $n$ samples from $P^{\mathrm{true}} \equiv \mathcal{L}_2 \in \mathcal{P}$. The hypotheses space $\mathcal{H}$ is defined as a finite set of simple binary classifiers containing at least one realizable hypothesis, and we consider the zero-one loss $l$ as we did in the main portion of the paper.

We need to find $\cup_{P^{\mathrm{ex}} \in \mathrm{ex}\mathcal{P}} B_{P^{\mathrm{ex}}} = \cup_{i=1}^3 \{h \in \mathcal{H} : L_{\mathcal{L}_i}(h) > \epsilon\}$, where $\epsilon$ depends on $\delta$ as in the proof of Theorem 4.1. That is, we want those $h$'s for which the expected loss according to $\mathcal{L}_1$ or $\mathcal{L}_2$ or $\mathcal{L}_3$ is larger than $\epsilon$. They are the collection of "bad hypotheses" according to at least one of the extreme elements of our credal set. Recall that

$$\epsilon^\star(\delta) = \frac{\log|\cup_{P^{\mathrm{ex}} \in \mathrm{ex}\mathcal{P}} B_{P^{\mathrm{ex}}}| + \log(1/\delta)}{n}$$

is the bound we found in Theorem 4.1, and that

$$\epsilon_{\mathrm{UB}}(\delta) = \frac{\log|\mathcal{H}| + \log(1/\delta)}{n}$$

is the classical SLT bound, that we reported in Corollary 4.3.

As we can see from Table B.1, our bound $\epsilon^\star(\delta)$ improves on the classical SLT one $\epsilon_{\mathrm{UB}}(\delta)$. Table B.1 also tells us that bound $\epsilon^\star(\delta)$ is tighter than $\epsilon_{\mathrm{UB}}(\delta)$ when the sample size $n = |D_4|$ is small, and then $\epsilon^\star(\delta)$ becomes progressively closer to $\epsilon_{\mathrm{UB}}(\delta)$ as $n = |D_4|$ increases. This same pattern is observed when the extrema of the credal set are closer to each other. Indeed, in Table B.2 we repeat the experiment and choose as extrema of $\mathcal{P}$ three measures whose pdf's are three Normals $\mathcal{N}(-0.1, 1)$,

$\mathcal{N}(0, 1)$, and $\mathcal{N}(0.1, 1)$.[10] The reason for this behavior is the following. With few available samples, that is, when $n = |D_4|$ is low, Credal Learning Theory is able to leverage the evidence encoded in the credal set, and hence to derive a tighter bound than classical Statistical Learning Theory. When the sample size is large, that is, when $n = |D_4|$ is high, the classical bound $\epsilon_{\mathrm{UB}}(\delta)$ itself is very small. This is because the amount of evidence available is large, and so $1/n \sum_{i=1}^{n} l((x_i, y_i), h)$ well approximates $\int_{\mathcal{X} \times \mathcal{Y}} l((x, y), h) P^{\mathrm{true}}(\mathrm{d}(x, y))$. In turn, since $\epsilon_{\mathrm{UB}}(\delta)$ is already very small, then the CLT bound $\epsilon^{\star}(\delta)$ we derive cannot improve greatly on it. Hence, their values are close together, despite $\epsilon^{\star}(\delta)$ being slightly tighter. The code for this experiment is available upon request.

| # Samples $n$ | $\epsilon^{\star}(\delta)$ | $\epsilon_{\mathrm{UB}}(\delta)$ | $\left| \cup_{P^{\mathrm{ex}} \in \mathrm{ex}\mathcal{P}} B_{P^{\mathrm{ex}}} \right|$ | $|\mathcal{H}|$ | Realizability |
|---|---|---|---|---|---|
| 10 | 0.74500 | 0.76009 | 86 | 100 | Yes |
| 100 | 0.07560 | 0.07600 | 96 | 100 | Yes |
| 200 | 0.03785 | 0.03800 | 97 | 100 | Yes |
| 300 | 0.02526 | 0.02533 | 98 | 100 | Yes |
| 400 | 0.01897 | 0.01900 | 99 | 100 | Yes |
| 500 | 0.01516 | 0.01520 | 98 | 100 | Yes |

Table B.1: Results of experimental evaluation of our bound tightness. Here the hypotheses space is such that $|\mathcal{H}| = 100$, and $\delta = 0.05$. The likelihood pdfs $\ell_1 = \mathcal{N}(-5, 1)$, $\ell_2 = \mathcal{N}(0, 1)$, and $\ell_3 = \mathcal{N}(5, 1)$.

| # Samples $n$ | $\epsilon^{\star}(\delta)$ | $\epsilon_{\mathrm{UB}}(\delta)$ | $\left| \cup_{P^{\mathrm{ex}} \in \mathrm{ex}\mathcal{P}} B_{P^{\mathrm{ex}}} \right|$ | $|\mathcal{H}|$ | Realizability |
|---|---|---|---|---|---|
| 10 | 0.73777 | 0.76009 | 80 | 100 | Yes |
| 100 | 0.07549 | 0.07600 | 95 | 100 | Yes |
| 200 | 0.03780 | 0.03800 | 96 | 100 | Yes |
| 300 | 0.02520 | 0.02533 | 96 | 100 | Yes |
| 400 | 0.01892 | 0.01900 | 97 | 100 | Yes |
| 500 | 0.01514 | 0.01520 | 97 | 100 | Yes |

Table B.2: Results of experimental evaluation of our bound tightness. Here the hypotheses space is such that $|\mathcal{H}| = 100$, and $\delta = 0.05$. The likelihood pdfs $\ell_1 = \mathcal{N}(-0.1, 1)$, $\ell_2 = \mathcal{N}(0, 1)$, and $\ell_3 = \mathcal{N}(0.1, 1)$.

**Experiment 2:** We conducted another synthetic experiment to show that the empirical risk for a given distribution is upper bounded by the traditional SLT bound of Corollary 4.3. This is a sanity check to see whether the environment we used in Experiment 1 is a valid one to check our results.

For the experiment, we selected a standard Gaussian distribution $\mathcal{N}(0, 1)$ (mean 0, standard deviation 1) to generate data. Similarly to Experiment 1, (i) the hypotheses space $\mathcal{H}$ is defined as a finite set of simple binary classifiers containing at least one realizable hypothesis, and (ii) we assume a zero-one loss function. The latter is used to evaluate the performance of the classifiers. For each run, we generated a training set and a test set from the standard Gaussian distribution. At training time, for each hypothesis $h$ in $\mathcal{H}$, we calculate the empirical risk on the training set using the zero-one loss and identify the hypothesis $\hat{h}$ that minimizes such risk (the empirical risk minimizer). At test time, we compute the empirical risk $L_P(\hat{h})$ of $\hat{h}$ on the test set as shown in Table B.3.[11]

| Training Samples | Test Samples | $L_P(\hat{h})$ (Test) | $\epsilon^{\star}(\delta)$ (Test) | $\epsilon_{\mathrm{UB}}(\delta)$ (Test) |
|---|---|---|---|---|
| 1000 | 500 | 0.000 | 0.01514 | 0.01520 |
| 1500 | 1000 | 0.000 | 0.00757 | 0.00760 |
| 2000 | 1500 | 0.000 | 0.00506 | 0.00506 |

Table B.3: Results of experimental evaluation. The hypotheses space is such that $|\mathcal{H}| = 100$, and $\delta = 0.05$.

We calculate the upper bound $\epsilon_{\mathrm{UB}}(\delta)$ on the empirical risk based on Corollary 4.3, which is a function of the number of hypotheses in $\mathcal{H}$, the value of $\delta$, and the number $n$ of training samples. This is the

---

[10]Of course, they are much closer to each other than the Normals $\mathcal{N}(-5, 1), \mathcal{N}(0, 1)$, and $\mathcal{N}(5, 1)$, e.g. in the Total Variation metric.

[11]Here $P$ denotes the probability measure whose pdf is the standard Normal density.

classic SLT bound. The experiment is run 1000 times with specified numbers of training and test samples, and a $\delta$ value of 0.05 to check whether the condition (empirical risk on the test set is upper bounded by the theoretical bound) is satisfied in any of the 1000 trials. The experimental results in Table B.3 validate the classical SLT bound by repeatedly testing it on randomly generated data. The code for this experiment is also available upon request.

**Experiment 3:** In this, we aim to empirically validate Theorem 4.5, which addresses the behavior of the empirical risk minimizer in the presence of a finite hypothesis space and no realizability. We generate synthetic data from Gaussian distributions (with the same parameters as in Experiment 1), with added uniform noise to ensure no realizability, meaning that no hypothesis can perfectly predict the labels. The hypotheses space $\mathcal{H}$ is defined as a set of threshold-based classifiers parameterized by $\theta$. For the experiment, we generate training samples $D_1, D_2, D_3$ and test sample $D_4$ from Gaussian distributions with added noise. Labels are created based on the samples, with noise introduced to flip labels randomly, ensuring that no hypothesis in $\mathcal{H}$ can achieve zero loss. We identify the empirical risk minimizer $\hat{h}$ using the combined training samples $D_1, D_2, D_3$. We calculate the empirical risk $L_P(\hat{h})$ using the test data $D_4$. The theoretical risk $L_P(h^\star)$ is assumed to be the risk of a perfect classifier. We compute the theoretical bound $\epsilon^{\star\star}(\delta)$ and verify whether the difference $L_P(\hat{h}) - L_P(h^\star)$ is within this bound. The results show that the empirical risk of the empirical risk minimizer $\hat{h}$ is within the bound $\epsilon^{\star\star}(\delta)$ of the best theoretical model $h^\star$. The difference $L_P(\hat{h}) - L_P(h^\star)$ also satisfies the condition

$$L_P(\hat{h}) - L_P(h^\star) \leq \epsilon^{\star\star}(\delta).$$

This validates empirically (in a synthetic environment) Theorem 4.5, showing that even under no realizability, the empirical risk minimizer's performance is close to the theoretical best within a computable bound. The experimental results are presented in Table B.4, where we also show (i) that $\epsilon^{\star\star}(\delta) \leq \epsilon'_{\text{UB}}(\delta) \doteq \sqrt{\frac{2\left(\log|\mathcal{H}| + \log\left(\frac{2}{\delta}\right)\right)}{n}}$ from Corollary 4.7 always holds; and (ii) that by foregoing realizability, we obtain a slightly looser bound. Indeed, as we can see, $\epsilon^{\star\star}(\delta)$ is slightly larger than $\epsilon^\star(\delta)$ from Table B.1 for all the sample size values $n$ that we consider. The code for this experiment is also available upon request.

| # Samples $n$ | $L_P(\hat{h})$ | $L_P(h^\star)$ | $L_P(\hat{h}) - L_P(h^\star)$ | $\epsilon^{\star\star}(\delta)$ | $\epsilon'_{\text{UB}}(\delta)$ |
|---|---|---|---|---|---|
| 10 | 0.30000 | 0.10000 | 0.19999 | 0.76009 | 0.84877 |
| 100 | 0.14000 | 0.10000 | 0.04000 | 0.07600 | 0.26840 |
| 200 | 0.10500 | 0.10000 | 0.00499 | 0.03800 | 0.18979 |
| 300 | 0.11333 | 0.10000 | 0.01333 | 0.02533 | 0.15496 |
| 400 | 0.11750 | 0.10000 | 0.01749 | 0.01900 | 0.13420 |
| 500 | 0.10400 | 0.10000 | 0.00399 | 0.01520 | 0.12003 |

Table B.4: Results of experimental evaluation of Theorem 4.5. Here the hypotheses space is such that $|\mathcal{H}| = 100$, $\delta = 0.05$ and noise level 0.1. The likelihood pdfs $\ell_1 = \mathcal{N}(-5, 1)$, $\ell_2 = \mathcal{N}(0, 1)$, and $\ell_3 = \mathcal{N}(5, 1)$. Let us remark that in this experiment *we forego the assumption of realizability*, that $L_P(\hat{h}) - L_P(h^\star) \leq \epsilon^{\star\star}(\delta)$ *for every sample size we tested on*, and that $\epsilon^{\star\star}(\delta) \leq \epsilon'_{\text{UB}}(\delta)$ *for every sample size we tested on*.

# C   A Simple Numerical Example for the $\epsilon$-Contamination Model of Section 3.1.1

Just like in Section 3.1.3, let $\Omega = \{\omega_1, \omega_2, \omega_3\}$, where $\omega_j = (x_j, y_j)$, $j \in \{1, 2, 3\}$. Suppose also that we observed four training samples $D_1, \ldots, D_4$ and that we specified the likelihoods $\mathcal{L}_1, \ldots, \mathcal{L}_4$ as in the following Table.

|  | $\{\omega_1\}$ | $\{\omega_2\}$ | $\{\omega_3\}$ |
|---|---|---|---|
| $\mathcal{L}_1$ | 0.3 | 0.1 | 0.6 |
| $\mathcal{L}_2$ | 0.4 | 0.2 | 0.4 |
| $\mathcal{L}_3$ | 0.1 | 0.8 | 0.1 |
| $\mathcal{L}_4$ | 0.15 | 0.7 | 0.15 |

Then, suppose that $\epsilon_1 = 0.2$, $\epsilon_2 = 0.3$, $\epsilon_3 = 0.1$, and $\epsilon_4 = 0.25$, so that $\mathscr{L}_1 = \{P : P = 0.8\mathcal{L}_1 + 0.2Q, \forall Q \in \Delta_\Omega\}$, $\mathscr{L}_2 = \{P : P = 0.7\mathcal{L}_2 + 0.3Q, \forall Q \in \Delta_\Omega\}$, $\mathscr{L}_3 = \{P : P = 0.9\mathcal{L}_3 + 0.1Q, \forall Q \in \Delta_\Omega\}$, and $\mathscr{L}_4 = \{P : P = 0.75\mathcal{L}_4 + 0.25Q, \forall Q \in \Delta_\Omega\}$. By Wasserman and Kadane [74, Example 3], we know that, if we put $\mathcal{P} = \mathrm{Conv}(\cup_{i=1}^4 \mathscr{L}_i)$, the following holds

$$\underline{P}(A) = \begin{cases} \min_{i \in \{1,\dots,4\}}(1 - \epsilon_i)\mathcal{L}_i(A), & \forall A \neq \Omega \\ 1, & \text{if } A = \Omega \end{cases}$$

and

$$\overline{P}(A) = \begin{cases} \max_{i \in \{1,\dots,4\}}(1 - \epsilon_i)\mathcal{L}_i(A) + \epsilon_i, & \forall A \neq \emptyset \\ 0 & \text{if } A = \emptyset \end{cases}.$$

Simple calculations, then, give us the following values

| | $\underline{P}$ | $\overline{P}$ |
|---|---|---|
| $\{\omega_1\}$ | 0.09 | 0.58 |
| $\{\omega_2\}$ | 0.08 | 0.82 |
| $\{\omega_3\}$ | 0.09 | 0.68 |
| $\{\omega_1, \omega_2\}$ | 0.32 | 0.91 |
| $\{\omega_2, \omega_3\}$ | 0.42 | 0.91 |
| $\{\omega_1, \omega_3\}$ | 0.18 | 0.92 |
| $\{\omega_1, \omega_2, \omega_3\}$ | 1 | 1 |

As we can see, in this example too, the probability bounds imposed by the credal set are not too stringent, and in line with the evidence encapsulated in $\mathcal{L}_1, \dots, \mathcal{L}_4$. Hence, the assumption that $P^{\text{true}} \equiv P_5^\star \in \mathcal{P}$ is very plausible. For a visual representation of the credal set $\mathcal{P} = \mathrm{Conv}(\cup_{i=1}^4 \mathscr{L}_i)$, see the yellow convex region in the next figure; it is very similar to the convex region in Figure 2. This is unsurprising since the evidence used to derive the credal set in Section 3.1.3 is the same that we use to elicit $\mathcal{P}$ here.

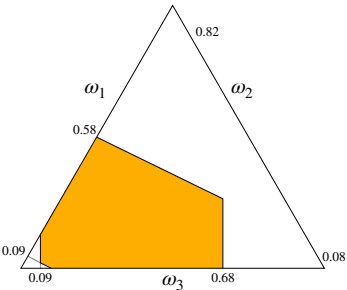

## D  A Fiducial Approach to Objectivist Modeling

An alternative objectivist approach to the ones presented in Section 3.1, proposed by Dempster and Almond [3], is based on *fiducial inference* [32]. Consider a parametric model, i.e., a family of conditional probability distributions of the data $\{f(\omega|\theta) : \omega \in \Omega, \theta \in \Theta\}$, where $\Omega$ is, again, the observation space and $\Theta$ is a parameter space. If the parametric (sampling) model is supplemented by a suitably designed auxiliary equation $\omega = a(\theta, u)$, where $u$ is a "pivot" variable of known a-priori distribution $\mu$, one obtains a random set $\Gamma$ mapping pivot values $u$ to subsets

$$\Gamma(u) = \{(\omega, \theta) \in \Omega \times \Theta : \omega = a(\theta, u)\}$$

of $\Omega \times \Theta$. This, in turn, induces a belief function on the product space $\Omega \times \Theta$ defined as

$$\mathrm{Bel}(A) = \sum_{u \in U : \Gamma(u) \subset A} \mu(u), \quad A \subset \Omega \times \Theta.$$

This can be finally be marginalized to the data space $\Omega = \mathcal{X} \times \mathcal{Y}$ to generate a belief function there. This approach was further extended by Martin, Zhang, and Liu, who used a "predictive" random set to express uncertainty on the pivot variable itself, leading to a *weak belief* inference technique [78].

In our framework, in which a finite sample of $N$ training sets $\{D_i\}_{i=1}^N$ is available, we can derive $N$ many random sets $\Gamma_i$ as before, $i \in \{1, \ldots, N\}$, and consider the $N$ belief functions $\mathrm{Bel}_i$ on $\Omega \times \Theta$ they induce. Then, we can compute their marginalization $\mathrm{Bel}_i|_\Omega$ on the data space $\Omega = \mathcal{X} \times \mathcal{Y}$, and compute the minimum $\underline{\mathrm{Bel}}|_\Omega \doteq \min_{i \in \{1, \ldots, N\}} \mathrm{Bel}_i|_\Omega$. It is easy to see that $\underline{\mathrm{Bel}}|_\Omega$ is itself a well-defined belief function. Finally, our credal set is given by $\mathcal{P} = \mathcal{M}(\underline{\mathrm{Bel}}|_\Omega)$ as in Section 3.1.2.

## E  Related Work on the Computational Complexity of Credal Sets

In this section, we discuss some of the analyses existing in the literature of the computational complexity specific to the use of credal sets, particularly in the context of graphical models and probabilistic inference. Such approaches can be implemented for large datasets, but they often require approximation techniques to be computationally feasible [45, 53, 54]. Despite this, credal set approaches can be implemented for large datasets using techniques like parallel processing, distributed computing, and efficient data structures. Similar to Deep Learning-based approaches, utilization of high-performance computing resources, algorithm optimization, and domain-specific adaptations, the computational challenges can be effectively managed. Recent advancements demonstrate the practicality of these approaches. For instance, Credal-Set Interval Neural Networks (CreINNs) have shown significant improvements in inference time over variational Bayesian neural networks [73]. Thus, while the computational demands are comparable to those of deep learning-based methods, the robustness and flexibility of credal sets, as demonstrated in recent research, make them a practical and valuable approach [51, 72].

## F  On the Relation Between Credal Sets and Continual Learning

The credal approach in this paper is closely linked to Continual Learning applications, which emphasize the need to handle diverse and sequential datasets to achieve robust and generalizable models. Recent works in continual learning have demonstrated the practical applications and benefits of using a multi-dataset setup. For instance, Jeeveswaran et al. [37] introduce a novel method for domain incremental learning, leveraging multiple datasets to adapt seamlessly across different tasks. Another example is Yu et al. [77], who propose a parameter-efficient continual learning framework that dynamically expands a pre-trained CLIP model through Mixture-of-Experts (MoE) adapters in response to new tasks. Ye et al. [76] address the challenges of multi-modal medical data representation learning through a continual self-supervised learning approach. These examples from recent studies demonstrate the practical applications and benefits of using a multi-dataset setup in a continual learning framework. Furthermore, some techniques use a multi-dataset setup in continual learning without relying on a specific temporal order. For example, Alssum et al. [4] present a replay mechanism based on single frames, arguing that video diversity is more crucial than temporal information under extreme memory constraints. By storing individual frames rather than contiguous sequences, they can maintain higher diversity in the replay memory, which leads to better performance in continual learning scenarios.

## NeurIPS Paper Checklist

The checklist is designed to encourage best practices for responsible machine learning research, addressing issues of reproducibility, transparency, research ethics, and societal impact. Do not remove the checklist: **The papers not including the checklist will be desk rejected.** The checklist should follow the references and precede the (optional) supplemental material. The checklist does NOT count towards the page limit.

Please read the checklist guidelines carefully for information on how to answer these questions. For each question in the checklist:

- You should answer [Yes] , [No] , or [NA] .
- [NA] means either that the question is Not Applicable for that particular paper or the relevant information is Not Available.
- Please provide a short (1–2 sentence) justification right after your answer (even for NA).

**The checklist answers are an integral part of your paper submission.** They are visible to the reviewers, area chairs, senior area chairs, and ethics reviewers. You will be asked to also include it (after eventual revisions) with the final version of your paper, and its final version will be published with the paper.

The reviewers of your paper will be asked to use the checklist as one of the factors in their evaluation. While "[Yes] " is generally preferable to "[No] ", it is perfectly acceptable to answer "[No] " provided a proper justification is given (e.g., "error bars are not reported because it would be too computationally expensive" or "we were unable to find the license for the dataset we used"). In general, answering "[No] " or "[NA] " is not grounds for rejection. While the questions are phrased in a binary way, we acknowledge that the true answer is often more nuanced, so please just use your best judgment and write a justification to elaborate. All supporting evidence can appear either in the main paper or the supplemental material, provided in appendix. If you answer [Yes] to a question, in the justification please point to the section(s) where related material for the question can be found.

IMPORTANT, please:

- **Delete this instruction block, but keep the section heading "NeurIPS paper checklist",**
- **Keep the checklist subsection headings, questions/answers and guidelines below.**
- **Do not modify the questions and only use the provided macros for your answers**.

