# OpenReview forum: "Credal Learning Theory"
_NeurIPS.cc/2024/Conference — NeurIPS 2024 poster_

### Official Review · Reviewer_SbDN · 2024-07-04

**Soundness:** 3
**Presentation:** 2
**Contribution:** 2
**Rating:** 6
**Confidence:** 2

**Summary:**

This paper extended traditional statistical learning theory which usually considers a fixed underlying data-generating distribution to a case where the underlying distribution is assumed to be from a convex set of distributions. Several excess risk bounds (finite realizable, finite unrealizable, and infinite hypothesis space) were derived.

**Strengths:**

- This paper aims to analyze the uncertainty and divination of training/test data distributions in a principled manner, which is very appreciated.
- Using convex sets of distributions is a reasonable approach.
- This paper considered different assumptions on the hypotheses.

**Weaknesses:**

- The writing can be improved. For example, the first paragraph is basically the notation and a review of ERM. The second paragraph mentioned DA and DG, but it is still unclear what problems existing theories and problem settings have and why it is important to solve them. Then, the author implied that the existing assumptions are unrealistic and too strong so the resulting theories are not generalizable. However, these statements are too vague so it is difficult to verify or falsify them.
- Partly due to the writing, I cannot fully understand the proposed theorems, let alone their proofs. For example, I do not understand what exactly a _well-defined_ quantity $\epsilon^\star(\delta)$ in Theorem 4.1 (and the following theorems) is. Therefore, the theoretical implications are very unclear to me, and I cannot accurately assess the significance and novelty of this work.

**Questions:**

Minor issues:
- ERM, DA, DG: please provide references.
- the position of Figure 1.
- l. 74: the citing style "Zhou et al. [63]" seems strange (to me)

**Limitations:**

The author discussed the limitations briefly in the conclusion section.

---

> ### Author Rebuttal · Authors · 2024-08-06
>
> *The writing can be improved. For example, the first paragraph is basically the notation and a review of ERM.*
>
> We thank the reviewer for their input. In the updated version, we will strive to further improve our writing. We believe, however, that the first paragraph is needed to set up the notations, so that every reader can be put in a position to follow our arguments, regardless of their background knowledge of the matter.
>
> *The second paragraph mentioned DA and DG, but it is still unclear what problems existing theories and problem settings have and why it is important to solve them. Then, the author implied that the existing assumptions are unrealistic and too strong so the resulting theories are not generalizable. However, these statements are too vague so it is difficult to verify or falsify them.*
>
> Regarding DA and DG, we agree with the reviewer, and we will improve clarity regarding existing theories and their shortcomings. At a very high level, no existing technique takes into account possible distribution misspecification and drift like we do, thanks to our credal set approach.
>
> Additional clarifications statements have been added in the Related Work section. The following changes have been made addressing the reviewer’s comments:
>
>
> In line 78, remove from ", or are reliant on strong assumptions (e.g.," until line 80 up to "DG approaches." Include:
> "Regarding Kernel-based methods, assumptions related to boundedness of Kernels and continuity of feature maps and loss function render the approaches not directly applicable to broader scenarios (Deshmukh et al., 2019; Hu et al., 2020; Ye et al., 2021)"
>
>
>
>
> In line 82, after the sentence ending "H-divergence [1 ].", remove text up to line 84 ending in "robust generalization [49 ]." Include:
> Researchers have also focused on adaptation to new domains over time, treating DG as an online game and the model as a player minimizing the risk associated with introducing new distributions by an adversary at each step (Rosenfeld et al. , 2022).
> However, in scenarios where the training distribution is significantly outside the convex hull of training distributions (Albuquerque et al. 2019), or because of unmet strong convexity loss function  assumption (Rosenfeld et al., 2022), they fall short from achieving robust generalization.
>
>
> Remove text from line 90 up to line 92 ending in "an adversary at each step [49 ]."
>
>
> After line 96 ending in "range of models [ 36 ]. ", include:
> Though this simplification has a number of practical benefits, models trained under covariate shift assumptions might suffer in terms of robustness to other distribution shift types.
>
>
> *Partly due to the writing, I cannot fully understand the proposed theorems, let alone their proofs. For example, I do not understand what exactly a well-defined quantity $\epsilon^\star(\delta)$ in Theorem 4.1 (and the following theorems) is. Therefore, the theoretical implications are very unclear to me, and I cannot accurately assess the significance and novelty of this work.*
>
> We thank the reviewer for pointing this out. Because of the 9 page limitation, we were not able to deliver the proofs to our results in the main part of the paper. They are postponed to Appendix A. We will likely not be able to move them to the main body of the paper, as there is a limitation in the rebuttal phase as well. The interested reader, though, can find there proofs to all the statements in the appendix mentioned.
>
> We note in passing that we use the term "well defined" in the usual mathematical sense of "expression whose definition assigns it a unique interpretation or value" [1]. In particular, given any value of $\delta$, the quantity $\epsilon^\star(\delta)$, depending on $\delta$, is indeed well-defined. To see this, we refer the reviewer to the last equation on page 13.
>
> [1] Joseph A. Gallian, Contemporary Abstract Algebra, Houghton Mifflin, 2006.
>
> *ERM, DA, DG: please provide references*
>
> We thank the reviewer for pointing this out. Regarding ERM, we refer to Liang’s lecture notes and [38] extensively in our manuscript. In the new version of the manuscript, we will mention them already in the first page. Regarding DG, we explicitly mention [47] in the last line of page 1. We will add a reference for DA in the new version (https://link.springer.com/article/10.1007/s10994-009-5152-4).
>
> *the position of Figure 1.*
>
> We thank the reviewer for their suggestion, and we will move Figure 1 right before the "Paper outline" subsection of the paper.
>
> *l. 74: the citing style "Zhou et al. [63]" seems strange (to me)*
>
> We do not know how to handle this suggestion, we must confess. It is the visualization of the \citet command that we use extensively throughout the paper.

---

> > ### Comment · Reviewer_SbDN · 2024-08-12
> >
> > Thank you for your detailed reply. I've read other reviews and responses. I'm now more positive about this paper but still not so sure about my understanding, so I raised my score while keeping my confidence.
> >
> > My previous rating was mainly due to my insufficient understanding of this work. I appreciate the author's explanation of the problem setting, their technical contributions, and the practical relevance (e.g., the availability of the credal set, continual learning, distribution misspecification).
> >
> > I do not doubt that it is difficult to explain such a topic clearly to every reader in 9 pages, as I also faced similar issues before. The current presentation may be sufficient for some researchers, but as someone who only vaguely remembers those important results in statistical learning theory, I have to agree with `Reviewer hWdx` and `Reviewer 2ZTx` that this paper is quite dense. Maybe a 10-page conference paper is not the best form to present this theory. I'm looking forward to the author's other expositions of this theory (journal papers, lectures, tutorials, etc.), if possible, so that I may be able to use it in other settings.
> >
> > Regarding the citing style, do not worry and please ignore it. It's just a personal preference.

---

> > > ### Author Response · Authors · 2024-08-12
> > > **Thank you!**
> > >
> > > We thank the reviewer for the time they spent understanding more deeply our paper, for understanding our struggles with the page limit, and for raising their score. As they correctly suggest, we are currently preparing a journal version where we also plan to extend our results, and we are thrilled about their interest in possibly using our findings in their research.
> > >
> > > Once again, we thank the reviewer for their comments,
> > >
> > > The Authors

---

### Official Review · Reviewer_2ZTx · 2024-07-10

**Soundness:** 3
**Presentation:** 2
**Contribution:** 3
**Rating:** 6
**Confidence:** 3

**Summary:**

This is a paper on learning theory with a focus on machine learning. The authors consider a setup with several training sets available.  Given an additional (test) set, the authors assume that the distribution generating these new data coincides with one of the distributions generating the training set or is a convex combination of them.  Under a few additional assumptions, the authors can find bounds on the expected loss of the empirical risk minimiser.

**Strengths:**

The paper is quite technical and dense. Yet, the proofs gathered in the appendix are relatively easy to follow and, after a preliminary check, correct. I think the generalised setup considered by the authors is quite interesting and challenging.

The author also presents an experimental evaluation of their bounds. This is very valuable for such a kind of theoretical paper.

**Weaknesses:**

The setup considered by the authors might appear a bit special and not very common.

Most of the results are a generalisation to the multi-dataset setup of results included in Liang's Lecture Notes. In a sense, the work in this paper might appear as a (straightforward?) extension of those results to the case of multiple datasets.

It is not very clear whether, relaxing the realizability assumption (something adding a lot of realism to the modelling) might have a strong impact to the bounds.

**Questions:**

- In which sense are the results in the paper not a simple corollary of those in Liang's LNs?
- Is it possible to characterise the impact of the realizability assumption on the bounds? What about considering this case even in the experiments?
- Is it possible to advocate the multi-dataset setup considered by the authors better? Are there examples of real tasks that cope with such a situation?
I found the experiments very good for the paper, while the discussion about the credal set learning was not so crucial. What about having the latter in the appendix and the former in the main body of the paper?

**Limitations:**

The limitations discussed by the authors wrt the experiments are very reasonable and I see no problems with them.

---

> ### Author Rebuttal · Authors · 2024-08-06
>
> *In which sense are the results in the paper not a simple corollary of those in Liang's LNs?*
>
> We thank the reviewer for this deep question, and for giving us the opportunity to be clearer about this topic.
>
> Theorem 4.1 is an immediate extension (not a corollary though) to the credal case of Liang’s LNs, as shown in Corollary 4.3. Corollaries 4.2 and 4.4, instead, cannot be immediately traced back to Liang. The first because it shows that the diameter of the credal set plays a role in deriving a bound to the expected risk of the ERM computed according to any possible distribution within the credal set itself (not necessarily the oracle one). Corollary 4.4, instead, inspects the possibility of distribution drift in the newly observed dataset, something Liang does not take into consideration. Similar considerations hold for the results in Sections 4.2 and 4.3.
> Let us add two considerations. Theorem 4.5 utilizes a different proof technique than the analogous, non-credal counterpart in Liang’s work, and Theorem 4.9 requires the definition of the extremal Rademacher complexity $R_{n,P^{ex}}(\mathcal{A})$, which, to the best of our knowledge, has never been introduced before.
>
> *Is it possible to characterize the impact of the realizability assumption on the bounds? What about considering this case even in the experiments?*
>
> We thank the reviewer for this deep question. Yes, it is possible. By looking at the proofs of Theorem 4.1 (in particular the last equation of page 13) and Theorem 4.5 (in particular Equation (9)) in Appendix A, we see how foregoing realizability implies a slightly looser bound. We did not include this consideration in the original version of the manuscript because of page limitations, but we will do so in the updated version. A newly added synthetic experiment  shows this behavior. In the experiment we computed the theoretical bound $\epsilon^{\star\star}(\delta)$ and verify (i) that it is slightly looser (e.g. with 200 samples, $\epsilon^{\star}(\delta) = 0.03785$ while $\epsilon^{\star\star}(\delta) = 0.03800$) and (ii) whether the difference $L_P(\hat{h}) - L_P(h^\star)$ is within this bound. The results showed that the empirical risk of the empirical risk minimizer $\hat{h}$ is within the bound $\epsilon^{\star\star}(\delta)$ of the best theoretical model $h^\star$. The difference $L_P(\hat{h}) - L_P(h^\star)$ also satisfies the condition
> $$L_P(\hat{h}) - L_P(h^\star) \leq \epsilon^{\star\star}(\delta).$$
> The experiment validates empirically (in a synthetic environment) Theorem 4.5, showing that even under no realizability, the empirical risk minimizer's performance is close to the theoretical best within a computable bound. The experimental results are presented in revised manuscript’s Table B.4, and can be seen here: https://shorturl.at/x6JFq
>
> *Is it possible to advocate the multi-dataset setup considered by the authors better? Are there examples of real tasks that cope with such a situation?*
>
> Our approach is closely linked with continual learning applications, which emphasize the need to handle diverse and sequential datasets to achieve robust and generalizable models. Recent works in continual learning have demonstrated the practical applications and benefits of using a multi-dataset setup. For instance, [1] introduces a novel method for domain incremental learning, leveraging multiple datasets to adapt seamlessly across different tasks. Another example is [2], which proposes a parameter-efficient continual learning framework that dynamically expands a pre-trained CLIP model through Mixture-of-Experts (MoE) adapters in response to new tasks. [3] addresses the challenges of multi-modal medical data representation learning through a continual self-supervised learning approach. These examples from recent studies demonstrate the practical applications and benefits of using a multi-dataset setup in a continual learning framework. Furthermore, some techniques use a multi-dataset setup in continual learning without relying on a specific temporal order. For example, [4] presents a replay mechanism based on single frames, arguing that video diversity is more crucial than temporal information under extreme memory constraints. By storing individual frames rather than contiguous sequences, they can maintain higher diversity in the replay memory, which leads to better performance in continual learning scenarios.
>
> By including citations to these works and discussing their relevance, we will strengthen the advocacy for our approach in the updated version of our manuscript.
>
> References:
>
> [1] Jeeveswaran, K., et al. Gradual Divergence for Seamless Adaptation: A Novel Domain Incremental Learning Method. ICML 2024.
>
> [2] Yu, Jiazuo, et al. "Boosting continual learning of vision-language models via mixture-of-experts adapters." Proc. IEEE/CVF Conference on Computer Vision and Pattern Recognition (CVPR). 2024.
>
> [3] Ye, Y., et al. Continual Self-supervised Learning: Towards Universal Multi-modal Medical Data Representation Learning. " Proc. IEEE/CVF CVPR. 2024.
>
> [4] Alssum, Lama, et al. "Just a glimpse: Rethinking temporal information for video continual learning." Proc. IEEE/CVF CVPR. 2023.
>
> *I found the experiments very good for the paper, while the discussion about the credal set learning was not so crucial. What about having the latter in the appendix and the former in the main body of the paper?*
>
> We thank the reviewer for the suggestion, and we will try to move the experiments in the main body of the paper. The current structure is the result of feedback we received in the importance of eliciting actual credal sets from the available data. Because deriving credal sets directly from data is currently an open question in Imprecise Probabilistic Machine Learning, in the new version we plan to keep a sketch of both how to derive credal sets, and of the key experimental results, in the main part of the manuscript.

---

> > ### Comment · Reviewer_2ZTx · 2024-08-12
> >
> > I thank the reviewers for their detailed feedback and for the additional work. I am happy to confirm my initial positive opinion about the paper.

---

### Official Review · Reviewer_sJhN · 2024-07-12

**Soundness:** 3
**Presentation:** 3
**Contribution:** 2
**Rating:** 5
**Confidence:** 3

**Summary:**

This paper introduces a novel learning framework termed “Credal Learning Theory” which extends the traditional statistical learning theory to handle variations in data distributions, especially the topic “domain generalization”. The authors propose using credal sets, which are convex sets of probability distributions, to model the variability in data generating processes. By leveraging multiple training sets generated from different distributions, the framework infers a credit set that allows the model to capture uncertainty and have guarantees in domain generalization. The paper also provides theoretical bounds on the expected risk of models learned within this credal framework under various conditions, including finite and infinite hypothesis spaces, and both realizable and non-realizable scenarios.

**Strengths:**

This paper introduces the novel concept of Credal Learning Theory to account for data distribution variability using credal sets. This approach presents a new perspective for analyzing the domain generalization field. The authors innovatively apply techniques from Imprecise Probabilities to infer models from limited multiple sample sets, which is practically significant. The definition of credal sets as the convex closure of a family of distributions is also quite natural. Additionally, the paper provides complete mathematical proofs to support the derived generalization bounds. The use of credal sets to model epistemic uncertainty is robust and well-justified, providing theoretical guarantees.

**Weaknesses:**

My main concern is whether the use of the credal set introduced in this paper is sufficient for analyzing domain generalization. As the paper describes, the model is inferred from a finite number of datasets, and we are studying cases where each dataset is sampled from one of a potential family of distributions. Domain generalization aims to provide guarantees for generalization across different distributions within this potential family. The paper provides generalization guarantees for all distributions within the credal set constructed by convex combinations of inferred distributions. However, the gap lies in whether the credal set adequately represents the true potential family of distributions. To convincingly address this, the paper should at least consider the following two aspects:

1. Provide real-world examples demonstrating that the constructed credal set encompasses most of the potential distributions we need to consider in these scenarios, thus illustrating the practical relevance of the credal set.
2. Make assumptions about the potential family of distributions, such as assuming the family follows a certain distribution, and prove that the credal set occupies a significant proportion of the support region with high density for this distribution. This would show that the generalization guarantees given for the credal set are meaningful for real-world distribution scenarios.

These two points are essential to demonstrate the universality of the credal set. Otherwise, there may be cases where the distributions of multiple datasets used for inferring the model are very close to each other, making their convex combination very small and only covering a tiny part of the support of the true family of distributions. This would mean there is no guarantee for the true potential family of distributions.

**Questions:**

Please refer to Weaknesses. In addition to the issues mentioned in Weaknesses above, are there any analyses of Computational Complexity that are specific to the use of credal sets as an approach to infer models? Can such an approach be implemented for limited but large datasets?

**Limitations:**

The authors have discussed the limitation of this paper in Section 5.

---

> ### Author Rebuttal · Authors · 2024-08-06
>
> *Provide real-world examples demonstrating that the constructed credal set encompasses most of the potential distributions we need to consider in these scenarios, thus illustrating the practical relevance of the credal set.*
>
> We thank the reviewer for the question, and for giving us the opportunity to expand on the matter. In general, as the reviewer points out and as we mention in our manuscript, a general method to guarantee (even probabilistically) that the true distribution of the new dataset will be included in the credal set is still to be developed. That being said, in many applications – e.g. to continual learning (CL) – scholars often make assumptions that make the use of credal sets with suitable coverage properties plausible. Think for example of the task similarity assumption in CL https://shorturl.at/UJuiZ. There, it is posited that the oracle distributions pertaining to all the tasks of interest are all contained in a TV-ball of radius $r$ chosen by the user. This captures the idea that the model studied will be used on tasks that are not too different from each other. In this case, the credal set elicited from a finite sample of distributions from the ball will be a good approximation of the ball itself, converging for $N \rightarrow \infty$ to the entire ball. In the healthcare setting, experts’ opinions can be incorporated alongside empirical data (plausible probability distributions) to represent the probability uncertainty, for example, for the prognosis of a disease given a set of patient characteristics/biomarkers (see e.g. https://shorturl.at/CTKid). To make sure that the credal set constructed encapsulates most of the `potential distributions needed’, a number of approaches can be taken including incremental learning; in this approach the AI model learns and updates knowledge incrementally. As a result, the credal set can be continuously updated (via incremental learning) as new data become available. In this direction, learning health systems are being implemented in practice. These are health systems “in which internal data and experience are systematically integrated with external evidence, and that knowledge is put into practice” https://shorturl.at/ubSCG. Once again, in the future we will study how to guarantee that the true distribution is an element of the credal set in full generality, and we will apply our findings to real-world datasets.
>
> *Make assumptions about the potential family of distributions, such as assuming the family follows a certain distribution, and prove that the credal set occupies a significant proportion of the support region with high density for this distribution. This would show that the generalization guarantees given for the credal set are meaningful for real-world distribution scenarios.*
>
> We thank the reviewer for their point, which ties to the one in the previous question. They are right in pointing out that – given that no general way (yet) exists of guaranteeing that the oracle distribution for the $(N+1)$-th dataset belongs to the credal set obtained from the previous $N$ ones (as also pointed out e.g. in Section 7 of https://shorturl.at/zbXiu) – assumptions must be made on a case-by-case basis. Once those assumptions are made, then it should be shown that either the credal set covers a non-negligible portion of the distribution class of interest, or that even a small credal set is "good enough". We will make this clear in the new version of the manuscript.
>
> We note in passing, though, that since our goal is maximal generality, the methods that we presented in Section 3 induce rather large credal sets. Consider for example Section 3.1.1. There, we first $\epsilon$-perturb all the likelihoods that we elicit for the $N$ "past" datasets, and then we take the convex hull of the union of such perturbations. If the $N$ likelihoods are "diverse enough", we will be able to cover a "wide area" of the distribution class of interest. For example, imagine that the class is "univariate continuous distributions supported on $\mathbb{R}$". If $\mathcal{L}_1$ is a Normal centered at $\mu_1$, and $\mathcal{L}_2$ is a Normal centered at $\mu_2$, $\mu_1 \neq \mu_2$ and "far enough" from each other, both having same (or similar) variance, then by considering the credal set induced by these two, we already cover virtually all distributions in the class. To see this, notice that, as pointed out in the equation between lines 145 and 146, the credal set includes all distributions that setwise dominate the lower probability. In addition, by Example 3 in https://tinyurl.com/3ftw7hrn, we know that the lower probability of any set $A$ in the sigma algebra is given by $\min_i (1-\epsilon_i) \mathcal{L}_i (A)$. Now, since the tails of a Normal distribution decay fastly, this means that  $\min_i (1-\epsilon_i) \mathcal{L}_i (A)$ is very close to $0$, for all $A$. To visualize this, please refer to the second picture in Example 1 of https://shorturl.at/x0VbJ. There, we have two Normals depicted, one in red and one in blue, having the same variance, the former (call it $L_1$) centered at 9, and the latter (call it $L_2$) centered at 20. As we can see,  $\min_i \mathcal{L}_i$ is basically zero everywhere, and in turn so will be $\min_i (1-\epsilon_i) \mathcal{L}_i$, for any $\epsilon_i >0$ that we choose. But then, almost all univariate continuous distributions on $\mathbb{R}$ will setwise dominate such a lower probability, resulting in a "virtually all-encompassing" credal set.
>
> *Are there any analyses of Computational Complexity that are specific to the use of credal sets as an approach to infer models? Can such an approach be implemented for limited but large datasets?*
>
> We thank the reviewer for their question, which we will answer in the Global Rebuttal.

---

> > ### Comment · Reviewer_sJhN · 2024-08-09
> >
> > Thanks for your response.

---

> > > ### Author Response · Authors · 2024-08-09
> > > **Thank you!**
> > >
> > > If our answers contributed to solve the doubts that the reviewer had (which we strongly hope), we would greatly appreciate it they could raise their score.
> > >
> > > Sincerely,
> > >
> > > The Authors

---

> ### Author Response · Authors · 2024-08-06
> **Last Question**
>
> **We report here too our answer to Reviewer sJhN's last question**
>
> *Are there any analyses of Computational Complexity that are specific to the use of credal sets as an approach to infer models? Can such an approach be implemented for limited but large datasets?*
>
> We thank the reviewer for their question, the answer to which we will try to add to the new version of our manuscript. Yes, there are analyses in the literature of computational complexity specific to the use of credal sets, particularly in the context of graphical models and probabilistic inference. Such approaches can be implemented for large datasets, but they often require approximation techniques to be computationally feasible [1, 2, 3 ].
> Despite this, credal set approaches can be implemented for large datasets using techniques like parallel processing, distributed computing, and efficient data structures. Similar to Deep Learning-based approaches, utilization of high-performance computing resources, algorithm optimization, and domain-specific adaptations, the computational challenges can be effectively managed. Recent advancements demonstrate the practicality of these approaches. For instance, "Credal-Set Interval Neural Networks" (CreINNs) have shown significant improvements in inference time over variational Bayesian neural networks [4]. Thus, while the computational demands are comparable to those of deep learning-based methods, the robustness and flexibility of credal sets, as demonstrated in recent research, make them a practical and valuable approach [5, 6].
>
> References:
>
> [1] Mauá, Denis Deratani, and Fabio Gagliardi Cozman. "Thirty years of credal networks: Specification, algorithms and complexity." International Journal of Approximate Reasoning 126 (2020): 133-157.
>
> [2] Lienen, Julian, and Eyke Hüllermeier. "Credal self-supervised learning." Advances in Neural Information Processing Systems 34 (2021): 14370-14382.
>
> [3] Mauá, Denis D., et al. "On the complexity of strong and epistemic credal networks." arXiv preprint arXiv:1309.6845 (2013).
>
> [4] Wang, Kaizheng, et al. "CreINNs: Credal-Set Interval Neural Networks for Uncertainty Estimation in Classification Tasks." arXiv preprint arXiv:2401.05043 (2024).
>
> [5] Marinescu, Radu, et al. "Credal marginal map." Advances in Neural Information Processing Systems 36 (2024).
>
> [6] Wang, Kaizheng, et al. "Credal Wrapper of Model Averaging for Uncertainty Estimation on Out-Of-Distribution Detection." arXiv preprint arXiv:2405.15047 (2024).

---

### Official Review · Reviewer_hWdx · 2024-07-12

**Soundness:** 3
**Presentation:** 2
**Contribution:** 3
**Rating:** 6
**Confidence:** 2

**Summary:**

The paper develops a so called credal learning theory that uses convex sets of probability distributions (also known as credal sets) to model the uncertainty of the data-generating distribution. As in the classical statistical learning theory, the paper derives new theoretical bounds on the risk of the models learned from a collection of datasets instead of a single dataset. These datasets do not necessarily correspond to the same distribution. The new results (i.e., bounds) can be viewed as a generalisation of the classical results.

**Strengths:**

- The paper looks at a core problem in machine learning and the credal sets based approach to bounding the expected risk of a machine learning model appears to be quite novel.

**Weaknesses:**

- In my opinion, the presentation is quite dense, the notation appears to be quite heavy in places and the paper is not easy to follow. I think it's important to illustrate some of the key concepts introduced in sections 3 and 4 with some examples. Tables 1 and 2 as well as Figure 2 are good but they seem somewhat disconnected from the rest of the paper.

**Questions:**

- In my understanding, the results derived in the paper rely on the availability of the credal set that is supposed to contain the true data generating process. How is that credal set elicited?

- In principle, the credal set can be arbitrarily large in the sense that it can have arbitrarily many extreme points. Is there a bound on its size?

**Limitations:**

The limitations of the proposed approach are clearly discussed in the paper.

---

> ### Author Rebuttal · Authors · 2024-08-06
>
> *In my opinion, the presentation is quite dense, the notation appears to be quite heavy in places and the paper is not easy to follow. I think it's important to illustrate some of the key concepts introduced in sections 3 and 4 with some examples. Tables 1 and 2 as well as Figure 2 are good but they seem somewhat disconnected from the rest of the paper.*
>
> We thank the reviewer for their insights. In the updated version, we will strive for more clarity. In the present version, we already have an example pertaining to Section 3.1.3, and we will try to bring to the main body of the paper the example pertaining to Section 3.1.1 that is currently confined in Appendix C (it was relegated there because of the 9 pages limitation).
> In fact, a synthetic example pertaining to section 4 can also be found in Appendix B; it was relegated there because of the page limitation. We will try to add a sketch of it in Section 4 to ease the way the paper reads.
>
> We will also try to be clearer about Tables 1 and 2. The former tells the reader, for example, that pmf $\ell_1$ assigns a probability of $0.3$ to the element $\omega_1$ of the state space $\Omega$, and similarly for the other pmf’s and the other elements of the state space. Table 2 tells the reader the values that the lower and upper probabilities assign to each element of the power set $2^\Omega$. They are computed according to [6, Section 4.4]. That is, for all $A\in 2^\Omega$,
>
> $$\underline{P}(A)=\max \left\lbrace{ \sum_{\omega \in A} \underline{P}(\omega) , 1 - \sum_{\omega \in A^c} \overline{P}(\omega)}\right\rbrace$$
>
> and
>
> $$\overline{P}(A)=1-\underline{P}(A^c)=\min \left\lbrace{ \sum_{\omega \in A} \overline{P}(\omega) , 1 - \sum_{\omega \in A^c} \underline{P}(\omega)}\right\rbrace.$$
>
> Finally, Figure 1 is a visual representation of the resulting credal set (focusing only on the singletons in the power set $2^\Omega$). As we can see, $\omega_3$ has a probability between $0$ and $0.75$, $\omega_1$ between $0$ and $0.5$, and $\omega_1$ between $0.25$ and $1$.
>
> *In my understanding, the results derived in the paper rely on the availability of the credal set that is supposed to contain the true data generating process. How is that credal set elicited?*
>
> We thank the reviewer for the opportunity to clarify on this point. Section 3 presents three ways of eliciting a credal set directly from the available data (see Sections 3.1.1, 3.1.2, 3.1.3, and 3.2), and an extra one is presented in Appendix D. Perhaps this was not clear enough, so we will strive for greater transparency in the new version.
>
> For the sake of completeness, let us summarize here the three approaches in the main portion of the paper. In the first one (Section 3.1.1) we first perturb the likelihood pertaining to each observed training set $D_i$, and then we take the convex hull of these perturbations. In the second one (Sections 3.1.2 and 3.1.3), we derive a plausibility function from the likelihoods, and we use the latter to characterize a credal set of probability measures whose probability density functions are pointwise dominated by the plausibility function. Finally, in Section 3.2 we illustrate the subjectivist approach, in which the scholar first specifies in a subjective manner – but influenced by the available empirical probabilities – the lower probability of some events of interest, then they extend (via Walley’s extension principle) these values to a lower probability defined over the whole power set $2^\Omega$, and in turn consider the credal set of probabilities that setwise dominate such an extended lower probability.
>
> *In principle, the credal set can be arbitrarily large in the sense that it can have arbitrarily many extreme points. Is there a bound on its size?*
>
> We thank the reviewer for this deep question; let us try to answer it. The size of the credal set does not necessarily depend on the number of its extreme elements. Think of a very small ball (whose extreme points are infinitely many) inscribed in a large polygon (having finitely many vertices, and hence finitely many extreme elements).
> Rather, one way of capturing the size of a credal set is to look at its diameter. To the best of our knowledge, there is no general recipe to bound it: it depends on how “far” (in some well-defined metric or divergence) the true data generating distributions for each of the collected training sets are. In this paper, we aimed at giving general results, in which one cannot control the diameter of the credal set. However, in many applications this may be indeed possible. In continual learning, for example, one can rely on the task similarity assumption (see e.g. Assumption 1 in https://arxiv.org/abs/2305.14782), that tells us that the data associated with each task are generated i.i.d. from distributions whose distance (e.g. in the TV metric) does not exceed some value $r$ specified by the user.

---

> > ### Comment · Reviewer_hWdx · 2024-08-12
> >
> > Thanks for the clarifications.

---

### Author Rebuttal · Authors · 2024-08-06

**We first answer to Reviewer sJhN's last question**

*Are there any analyses of Computational Complexity that are specific to the use of credal sets as an approach to infer models? Can such an approach be implemented for limited but large datasets?*

We thank the reviewer for their question, the answer to which we will try to add to the new version of our manuscript. Yes, there are analyses in the literature of computational complexity specific to the use of credal sets, particularly in the context of graphical models and probabilistic inference. Such approaches can be implemented for large datasets, but they often require approximation techniques to be computationally feasible [1, 2, 3 ].
Despite this, credal set approaches can be implemented for large datasets using techniques like parallel processing, distributed computing, and efficient data structures. Similar to Deep Learning-based approaches, utilization of high-performance computing resources, algorithm optimization, and domain-specific adaptations, the computational challenges can be effectively managed. Recent advancements demonstrate the practicality of these approaches. For instance, "Credal-Set Interval Neural Networks" (CreINNs) have shown significant improvements in inference time over variational Bayesian neural networks [4]. Thus, while the computational demands are comparable to those of deep learning-based methods, the robustness and flexibility of credal sets, as demonstrated in recent research, make them a practical and valuable approach [5, 6].

References:

[1] Mauá, Denis Deratani, and Fabio Gagliardi Cozman. "Thirty years of credal networks: Specification, algorithms and complexity." International Journal of Approximate Reasoning 126 (2020): 133-157.

[2] Lienen, Julian, and Eyke Hüllermeier. "Credal self-supervised learning." Advances in Neural Information Processing Systems 34 (2021): 14370-14382.

[3] Mauá, Denis D., et al. "On the complexity of strong and epistemic credal networks." arXiv preprint arXiv:1309.6845 (2013).

[4] Wang, Kaizheng, et al. "CreINNs: Credal-Set Interval Neural Networks for Uncertainty Estimation in Classification Tasks." arXiv preprint arXiv:2401.05043 (2024).

[5] Marinescu, Radu, et al. "Credal marginal map." Advances in Neural Information Processing Systems 36 (2024).

[6] Wang, Kaizheng, et al. "Credal Wrapper of Model Averaging for Uncertainty Estimation on Out-Of-Distribution Detection." arXiv preprint arXiv:2405.15047 (2024).

---

**We also attach a pdf with the Table pertaining to the new experiment we discussed in our answer to reviewer 2ZTx's second question.**

---

### Decision · Program_Chairs · 2024-09-25

**Decision:**

Accept (poster)

**Comment:**

This paper develops statistical learning bounds (e.g. PAC-style) based on the assumption that the user can identify a convex credal set that includes the true (test) distribution, and use this information to provide generalization guarantees. This allows the authors to derive bounds on the risk of models derived from multiple data sets, for example. Contributions are mainly theoretical, with some reviewers concerned about the density of notation and results.  I agree with one reviewer that the experiments (in the supplement) are helpful, but clearly the elicitation of the credal set is also critical, so there is some tension in how to allocate the available space.  Some reviewers were concerned about the practical implications, since the results depend on the correctness of the identified credal set, for example, but I do not consider this a significant flaw.